# Molecular Docking and Efficacy of *Aloe vera* Gel Based on Chitosan Nanoparticles against *Helicobacter pylori* and Its Antioxidant and Anti-Inflammatory Activities

**DOI:** 10.3390/polym14152994

**Published:** 2022-07-24

**Authors:** Reham Yahya, Aisha M. H. Al-Rajhi, Saleh Zaid Alzaid, Mohamed A. Al Abboud, Mohammed S. Almuhayawi, Soad K. Al Jaouni, Samy Selim, Khatib Sayeed Ismail, Tarek M. Abdelghany

**Affiliations:** 1Basic Sciences Department, College of Science and Health Professions, King Saud Bin Abdulaziz University for Health Sciences, Riyadh 11671, Saudi Arabia; yahyar@ksau-hs.edu.sa; 2King Abduallah International Medical Research Center, P.O. Box 3661, Riyadh 11481, Saudi Arabia; 3Department of Biology, College of Science, Princess Nourah bint Abdulrahman University, P.O. Box 84428, Riyadh 11671, Saudi Arabia; 4Assistant Agency for Preventive Health, Ministry of Health, Riyadh 11671, Saudi Arabia; szalzaid@moh.gov.sa; 5Biology Department Faculty of Science, Jazan University, Jazan 45142, Saudi Arabia; mohalabboud@hotmail.com (M.A.A.A.); kismail@jazanu.edu.sa (K.S.I.); 6Department of Medical Microbiology and Parasitology, Faculty of Medicine, King Abdulaziz University, Jeddah 21589, Saudi Arabia; 7Department of Hematology/Oncology, Yousef Abdulatif Jameel Scientific Chair of Prophetic Medicine Application, Faculty of Medicine, King Abdulaziz University, Jeddah 21589, Saudi Arabia; saljaouni@kau.edu.sa; 8Department of Clinical Laboratory Sciences, College of Applied Medical Sciences, Jouf University, Sakaka 72341, Saudi Arabia; 9Botany and Microbiology Department, Faculty of Science, Al-Azhar University, Cairo 71524, Egypt

**Keywords:** evaluation, in vitro, *Aloe vera*, chitosan nanoparticles, *Helicobacter pylor*, therapeutic effects

## Abstract

The medicinal administration of *Aloe vera* gel has become promising in pharmaceutical and cosmetic applications particularly with the development of the nanotechnology concept. Nowadays, effective *H. pylori* treatment is a global problem; therefore, the development of natural products with nanopolymers such as chitosan nanoparticles (CSNPs) could represent a novel strategy for the treatment of gastric infection of *H. pylori*. HPLC analysis of *A. vera* gel indicated the presence of chlorogenic acid as the main constituent (1637.09 µg/mL) with other compounds pyrocatechol (1637.09 µg/mL), catechin (1552.92 µg/mL), naringenin (528.78 µg/mL), rutin (194.39 µg/mL), quercetin (295.25 µg/mL), and cinnamic acid (37.50 µg/mL). CSNPs and *A. vera* gel incorporated with CSNPs were examined via TEM, indicating mean sizes of 83.46 nm and 36.54 nm, respectively. FTIR spectra showed various and different functional groups in CSNPs, *A. vera* gel, and *A. vera* gel incorporated with CSNPs. Two strains of *H. pylori* were inhibited using *A. vera* gel with inhibition zones of 16 and 16.5 mm, while *A. vera* gel incorporated with CSNPs exhibited the highest inhibition zones of 28 and 30 nm with resistant and sensitive strains, respectively. The minimal inhibitory concentration (MIC) was 15.62 and 3.9 µg/mL, while the minimal bactericidal concentration (MBC) was 15.60 and 7.8 µg/mL with MBC/MIC 1 and 2 indexes using *A. vera* gel and *A. vera* gel incorporated with CSNPs, respectively, against the resistance strain. DPPH Scavenging (%) of the antioxidant activity exhibited an IC_50_ of 138.82 μg/mL using *A.vera* gel extract, and 81.7 μg/mL when *A.vera* gel was incorporated with CSNPs. *A.vera* gel incorporated with CSNPs enhanced the hemolysis inhibition (%) compared to using *A.vera* gel alone. Molecular docking studies through the interaction of chlorogenic acid and pyrocatechol as the main components of *A. vera* gel and CSNPs with the crystal structure of the *H. pylori* (4HI0) protein supported the results of anti-*H. pylori* activity.

## 1. Introduction

In the last decade, natural products are becoming vital resources in numerous applications such as cosmetics industries, biopesticides, and the prevention and treatment of microbial and nonmicrobial diseases. As mentioned in numerous studies, *Aloe vera* is a tropical succulent plant that belongs to the liliaceous family; it is considered an important medicinal plant, which contains vital ingredients including amino acids, glycoproteins, phenolic compounds, polysaccharides, organic acids, lignin, hormones, vitamins, and saponins, as well as enzymes [1]. These components give this plant many important medicinal properties such as immune-boosting, antioxidant, anti-cancer, intestinal health promotion, anti-inflammatory, bone proliferation promotion, antimicrobial, neuroprotection, and hypoglycemic properties [2]. A strong anti-oxidative capacity of *A. vera* gel was reported previously due to the existence of various contents of phenolic compounds. The anti-oxidative activity reduces hydroperoxides, catalase, superoxide dismutase, and glutathione peroxidase [3]. The viscosity of *A. vera* gel may be due to its content of the sugar glucomannan, in addition to approximately 200 active components that were detected in *A. vera* gel among proteins and its monomers, lipids, vitamins, and polysaccharides [4]. The extracted gel of *A. vera* leaves exhibited a fungistatic activity against *Candida paraprilosis, Candida krusei*, and *C. albicans* [5,6]. As described previously, *Helicobacter pylori* inhabits the stomach of humans early in life; however, the related pathology may be expressed later. Surprisingly, 50% of the people worldwide are carriers of *H. pylori*, but infection symptoms appeared only in approximately 15%, with the increase in peptic ulcer, gastritis, and gastric adenocarcinoma [7]. Bacterial resistance, particularly *H. pylori*, to several antibiotics is considered a global problem for public health, and it has been of interest to many scientists to find a solution of this problem. Furthermore, significant challenges have emerged as a result of the failure to treat *H. pylori* infection; therefore, the discovery of a novel strategy combining nanoparticles and natural compounds has become a medical requirement for infection control.

*A. vera* inner gel was tested against *H. pylori* and the treatment of peptic ulcers [8]. Moreover, a synergistic efficacy was recorded in vitro using plant extracts with diverse antibiotics toward *H. pylori* strains [9]. Both susceptible and resistant *H. pylori* strains were influenced by *A. vera* gel according to a previous study [10].

Nanoparticles (NPs) have played and still play an important role in many medical, industrial, agricultural, environmental, and food fields [11,12,13,14,15,16,17,18,19,20,21,22,23]. From the applied NPs, chitosan nanoparticles (CSNPs) have been lately used at large scales for medical applications such as drug carriers and cosmetics [24].

The chemical structure of chitosan (cationic polymer) is composed of monomers of β-(1-4)-linked d-glucosamine and N-acetyl-d-glucosamine. The advantages of chitosan were recorded in the literature as biocompatible, nontoxic, bioavailable, and biodegradable; therefore, it was applied in wound healing, to inhibit the development of scar tissue, and for protein delivery, as well as other biomedical applications [25,26]. 

The synthesis of CSNPs has attracted the attention of many researchers due to the many applications of chitosan. Pharmaceuticals administration was reported in recent years using CSNPs due to their high activity level [27]. CSNPs are used as a drug carrier to support the medicinal features of drugs such as drug release regulation leading to drug solubility and stability enhancement. As a result of the incorporation of *A. vera* gel with CSNPs, the gel has gained several pharmacological properties that are of great interest to pharmaceutical fabrications [1]. The combination of CSNPs with *A. vera* extract exhibited an enhancement in wound healing caused by microbial pathogens [4]. Electrostatic attraction among cationic amino groups of chitosan with negative charges on the microbial cell wall is the main reason of microbial cell destruction. The activity of CSNPs loaded by *Satureja hortensis* essential oils had superior effects compared to CSNPs alone [28]. Other biological activities of *A. vera* gel were enhanced when it was incorporated with CSNPs such as their stability, in vitro release, and antioxidant potential [29]. Incorporation of *Byrsonima crassifolia* extract into CSNPs enhanced the control of some pathogenic fungi including *Colletotrichum gloeosporioides* and *Alternaria* species [30]. Phytoconstituents and its concentrations of *A. vera* as well as other plants may differ according to cultivation soil, climatic changes, type of fertilizers, and extraction methods. In addition, most of the studies on the incorporation of *A. vera* gel with CSNPs focused on wound healing and antimicrobial activity against certain microorganisms but not against *H. pylori*. Therefore, the aim of this study was to enhance the biological activities of *A. vera* gel with CSNPs as a natural and safe compound. We also studied the effects of this incorporation on *H. pylori* growth, antioxidant activity, and anti-inflammatory activity. Moreover, a docking study of the main components of *A. vera* gel on *H. pylori* was conducted. 

## 2. Materials and Methods

### 2.1. Chemicals Used

Chitosan nanoparticles (CSNPs) were purchased from Primex, Siglufjordur, Iceland. Other chemicals including buffers, reagents, solvents, and bacterial growth medium were obtained from Sigma-Aldrich (St. Louis, MS, USA). All chemicals used in the current experiments were of analytical grade. 

### 2.2. Plant Sample Used and Extraction Process

*Aloe vera* gel was collected from mature and healthy leaves to obtain 400 mL of slime of the gel through the cutting of leaves transversely into pieces. The gel was concentrated using a rotary evaporator at 50 °C. The final *A. vera* gel was concentrated to obtain 12 g. A voucher herbarium specimen had been deposited in the Botany and Microbiology Department, Faculty of Science, Al-Azhar University, Cairo, Egypt.

### 2.3. Flavonoid and Phenolic Contents Analysis by High-Performance Liquid Chromatography (HPLC) 

One gram of dried gel was extracted with 10 mL of methyl alcohol via a rotary evaporator to obtain concentrated extract. Five microliters of the extracted gel was injected in HPLC (Agilent 1260 series, Agilent Technologies, Santa Clara, CA, USA) characterized with the following brief conditions: The separation was performed using an Eclipse C18 column (4.6 mm × 250 mm i.d., 5 μm). Water and 0.05% trifluoroacetic acid in acetonitrile were applied as the mobile phase with a flow rate of 0.9 mL/min. The mobile phase consisted of water and 0.05% trifluoroacetic acid in acetonitrile (B) at a flow rate 0.9 mL/min, and the column temperature was adjusted at 40 °C. The mobile phase was programmed successively in a linear gradient with 60% at 0 min; 82% at 0–5, 5–8, 8–12, 12–15, 15–16, and 16–20 min. The UV-detector was set to 280 nm. The qualitative detection of phenolic and flavonoids was performed according to Abdelghany et al. [31] compared to the injected standards of phenolic and flavonoids in HPLC.

### 2.4. Preparation of Coating Solutions of CSNPs with Aloe vera Gel 

The CSNPs solution was prepared by dissolving chitosan in an aqueous solution (1%, *v*/*v*) of acetic acid, and then the addition of CSNPs obtained a final concentration of 2% (*w*/*v*). Glycerol was added to the prepared solution as a plasticizer at the concentration of 2% (*w*/*v*), followed by the addition of Tween 20 to the prepared solution at a concentration of 0.05% (*v*/*v*) in order to increase its wettability and adhesion properties. To obtain a CSNPs-*A. vera* gel composite coating solution, *A. vera* gel was incorporated into CSNPs solution (stirred for 25 min and then ultrasonicated for 45 min) to obtain the final concentration of *A. vera* gel concentrate/CSNPs in the solution at 10% by weight [32].

### 2.5. Characterization of CSNPs, Aloe vera Gel, and CSNPs Incorporated with Aloe vera Gel

#### 2.5.1. UV–Vis

The UV–vis absorption spectrum of the tested samples was scanned via a spectrophotometer (JASCO V-670, Thermo Fisher Scientific, Waltham, USA) at a range of 200–800 nm.

#### 2.5.2. Transmission Electron Microscopy 

The diameters and shape of CSNPs and CSNPs incorporated with *A. vera* gel were examined via Transmission Electron Microscopy (TEM) (JEOL-JEM-1200, JEOL, Tokyo, Japan). A drop of colloidal solution of each sample was loaded on copper grids (400 mesh) coated by amorphous carbon film. 

#### 2.5.3. Fourier Transform Infrared Spectroscopy

Fourier transform infrared spectroscopy (FTIR) (FT-IR; FTIR 8400S, Shimadzu, Tokyo, Japan) was applied to detect specific chemical groups of the tested samples. The CSNPs, dried *Aloe vera* gel extract, and CSNPs incorporated with *A. vera* gel were ground with KBr powder and pressed into pellets for FT-IR spectra measurement at a range of 400 to 4000 cm^−1^ of frequency.

### 2.6. Antimicrobial Activity of CSNPs, with Aloe vera Gel and CSNPs Incorporated with A. vera Gel

The agar diffusion method using Brain Heart Infusion medium with 7% serum was used to detect the activity of *A. vera* gel and *A. vera* gel incorporated with CSNPs against *H. pylori* (Hospital of Ain Shams University, Cairo, Egypt). Briefly, 100 μL of *H. pylori* suspension containing 10^8^ colony-forming units (CFUs)/mL was spread on the medium surface. Via a sterile cork borer with a diameter of 6 mm, a hole was punched from agar, and then 100 μL of the tested compounds at the desired quantity was introduced into the well. DMSO (solvent of gel) as a negative control while antibiotics including clarithromycin (CLR, 0.05 mg/mL), amoxicillin (AMX, 0.05 mg/mL), and metronidazole (MTZ, 0.8 mg/mL) as a positive control were applied. The plates were kept in a refrigerator for 30 min for appropriate diffusion of the tested compounds and control samples, and then transferred to an incubator for 3 days at 37 °C under microaerophilic conditions. At end of the incubation period, the diameter of clear zones around the well was measured [16,33].

### 2.7. Minimal Inhibitory Concentration and Minimal Bactericidal Concentration 

The minimal inhibitory concentration (MIC) of *A. vera* gel and *A. vera* gel incorporated with CSNPs was detected via the micro-dilution broth technique. The Mueller–Hinton (MH) broth provided with lysed horse blood was used for cultivation of *H. pylori*. Serial two-fold dilutions were performed to gain different concentrations ranging from 0.98 to 1000 μg/mL of the tested compounds. The sterile 96-well polystyrene microtitrate plates were prepared by distributing 200 μL of each dilution in broth medium per well. Fresh inocula of *H. pylori* were suspended in sterile NaCl (0.85%) to match the turbidity of 1.0 McFarland standard, and then 2 µL was transferred to the wells to attain 3.0 × 10^6^ colony forming units (CFU)/mL, followed by incubation under microaerophilic conditions with a limited amount of CO_2_ (15%) at 35 °C for 3 days. At the end of the incubation period, the MIC was assessed visually as the lowest concentration of each tested compound with the viewing of the complete inhibition of *H. pylori* growth. Microtitrate plates containing bacterial inoculum without the tested compounds (positive control) and tested compounds without bacterial inoculum (negative control) were applied in the current experiment. The minimal bactericidal concentration (MBC) was assayed by sub-culturing 100 mL of culture of the *H. pylori* from each well that showed thorough growth inhibition, from the last positive and from the growth control, onto the plates of MH agar provided with horse blood (5%), followed by incubation under microaerophilic conditions with a limited amount of CO_2_ (15%) at 35 °C for 3 days. MBC was determined at the lowest concentration of the tested compounds without the appearance of any *H. pylori* growth. The bactericidal or bacteriostatic effect of the tested compounds was evaluated via calculation of the MBC/MIC ratio. According to French [34], if the MBC/MIC ratio is no more than four times the MIC, the effect of the tested compounds is considered as bactericidal. Each trial was repeated in triplicate.

### 2.8. Antioxidant Activity

The antioxidant activity of *A. vera* gel and *A. vera* gel incorporated with CSNPs was evaluated via a 1,1-diphenyl-2-picryl hydrazyl (DPPH) radical scavenging assay. The free radical scavenging activity of different extracts of leaves plant were measured by 1,1-diphenyl-2-picryl hydrazyl (DPPH). One milliliter of 0.1 mM solution of DPPH in ethanol was mixed with 3 mL of the dissolved tested compounds in ethanol at various doses (3.9, 7.8, 15.62, 31.25, 62.5, 125, 250, 500, and 1000 μg/mL) that were prepared by sequential dilution. At room temperature (22 °C), the mixture was shaken for 30 min; then, via a spectrophotometer (UV-VIS milton roy), the absorbance of the reaction mixture was measured at 517 nm [31]. The dose of tested compounds necessary to inhibit 50% of the DPPH free radical (IC_50_) was recorded using a log dose inhibition curve. Ascorbic acid as a standard antioxidant agent was applied.
$$\mathrm{Efficacy}\;\mathrm{of}\;\mathrm{DPPH}\;\mathrm{scavenging}\;\left(\%\right)=\frac{\mathrm{CA}\;-\mathrm{TCA}\;\;}{\mathrm{CA}}\times 100$$
where CA and TCA are the absorbance of the control reaction and tested compound reaction, respectively.

### 2.9. Preparation of Erythrocyte Suspension and Hypotonicity-Induced Hemolysis

The collected blood sample (3 mL) from a healthy volunteer (corresponding author of current paper) was transferred to a heparinized tube, then centrifuged for 10 min at 3000 rpm. The obtained supernatant was mixed with an equivalent volume of normal saline that needed to dissolve the red blood pellets. The liquefied red blood pellets were reconstituted with isotonic sodium phosphate buffer (10 mM, pH 7.4) solution as 40/60% *v*/*v*. The used buffer consisted of 0.002 g of NaH_2_PO_4_, 0.016 g of Na_2_HPO_4_, and 0.09 g of NaCl per 100 mL of distilled water. The reconstituted red blood cells (re-suspended supernatant) were used as such.

The dissolved test compounds in distilled water represent the hypotonic solution. Different doses (100, 200, 400, 600, 800, and 1000 μg/mL) of tested compounds were added to 5 mL of hypotonic solution in centrifuge tubes. The same doses were added to 5 mL of isotonic solution in centrifuge tubes. Five milliliters of the vehicle (distilled water) and 5 mL of 200 μg/mL of indomethacin in each tube alone were used as a control. The suspended erythrocyte (0.1 mL) was added to each tube, followed by incubation at 37 °C for 60 min, then centrifuged for 3 min at 1300 g. Via a spectrophotometer (UV-VIS milton roy), the absorbance of the supernatant containing hemoglobin was estimated at 540 nm. The % of hemolysis was designed by supposing the hemolysis formed in the existence of distilled water as 100%. The % of hemolysis inhibition by the tested compounds was calculated:$$\mathrm{Inhibition}\;\mathrm{of}\;\mathrm{haemolysis}\;\left(\%\right)=1-\frac{\mathrm{OD}2-\mathrm{OD}1}{\mathrm{OD}3-\mathrm{OD}1}\times 100$$
where OD1 is the absorbance of tested compounds in isotonic solution; OD2 is the absorbance of tested compounds in hypotonic solution; OD3 is the absorbance of the control sample in hypotonic solution.

### 2.10. Molecular Docking

We studied the interaction between chlorogenic acid, chitosan, and pyrocatechol with the crystal structure of the *H. pylori* (4HI0) protein. A molecular modeling study using the Molecular Operating Environment (MOE) module was conducted to explain the observed antibacterial effect of the main detected compounds with the crystal structure of the *H. pylori* (4HI0) protein. MOE’s BUILDER module was used to create the structural model, and optimization conformational evaluations of the generated molecules were performed in two steps. The geometry of the compounds was optimized using the semiempirical PM3 Hamiltonian with the Restricted Hartree–Fock (RHF) and RMS gradient of 0.05 Kcal/mol, as well as the integrated MOPAC 7.0 energy minimization tool.

The resultant model was then used for the MOE’s ‘Systematic Conformational Search’. To rank the compounds’ binding affinity to the 4HI0 protein, the binding free energy and bonds of hydrogen among the compounds and amino acid into the 4HI0 protein were utilized. Estimation of the bonds of hydrogen was performed by determining the length of the hydrogen bond; moreover, the RMSD (Root-Mean-Square Deviation) of the co-crystal ligand position compared to the docking pose was utilized in ranking. Both RMSD as well as the mode of interaction of the native ligands within the crystal structure of the *H. pylori* (4HI0) protein receptor was utilized as a standard docked model.

### 2.11. Statistical Analysis

Assessments were performed in triplicate; therefore, it was estimated as mean ± standard deviation (SD). GraphPad Prism® software (version 5.0, GraphPad Software, San Diego, CA, U.S.) was applied to obtain the IC_50_ value of DPPH radical scavenging potential graphs.

## 3. Result and Discussion

### 3.1. Characterization of A. vera Gel and Phytochemical Analysis 

Two major parts were observed in the *A. vera* leaf including the outer green rind composed of vascular tissues and the interior colorless parenchyma tissues containing viscous clear liquid known as gel (Figure 1). Different terms were reported to describe the inner part of the *A. vera* leaf including the inner pulp, mucilaginous gel, inner gel, and mucilage tissue. According to previous investigators, approximately 99.5% water is the main content of the gel, while numerous contents represent 0.5–1% of the gel constituents [1]. 

Based on HPLC analysis, different flavonoids and phenolic acids were detected in the extract of *A. vera* gel with different concentrations and retention time (Table 1 and Figure 2). Chlorogenic acid was the main constituent followed by pyrocatechol and catechin with the concentrations of 1637.09, 1552.92, and 1100.43 µg/mL, respectively. Other compounds were recognized, namely naringenin (528.78 µg/mL), rutin (194.39 µg/mL), and quercetin (295.25 µg/mL) (Table 1). Cinnamic acid was also detected but with the lowest concentration (37.50 µg/mL). HPLC analysis of the *A. vera* gel extract showed the existence of 13 identified compounds in addition to unknowns with different areas (Figure 2), indicating its richness with phenolic and flavonoids contents. Our results agree with another study with some exceptions, for example, gallic acid and daidzein were detected in the *A. vera* gel under study but not detected in *A. vera* gel according to López et al. [35]. Guo and Mei [36] reported that the *A. vera* gel possesses diverse constituents and, therefore, it is applied in various medical purposes. The variation in concentration of these chemical constituents is based on the plant part used, extraction process, solvent, stage of growth, and plant source. All the detected compounds possess multiple biological functions in the medicinal fields as well as cosmetics and food industry. Our data match with some results by Hassan et al. [37] who observed that caffeic, coumaric, syringic, sinapic acid, cinnamic, and ferulic acid are the main acids of *A. vera* gel. Numan [38] confirmed the existence of phenolic compounds such as catechin, quercetin, aloe emodin, aloin, and sinapic acid in *A. vera* gel.

### 3.2. Characterization of CSNPs and CSNPs Incorporated with A. vera Gel

The UV–vis spectrum of CSNPs is shown in Figure 3. Broad absorption bands were observed at 270 nm. According to Thamilarasan et al. [39], the UV–Visible spectrum presented an absorption peak at 250 nm of CSNPs. Another absorption peak for CSNPs was obtained at 226 nm [40], while the absorption peak at 360 was observed with *A. vera* gel incorporated with CSNPs. TEM images indicated that the mean diameter of CSNPs was 83.46 nm with irregular shape, while the mean diameter of CSNPs incorporated with gel was 36.54 nm with similar forms of particles (Figure 4A,B). The diameters of some randomly selected CSNPs and CSNPs incorporated with gel were recorded (Figure 4C). The size of *A. vera* gel incorporated with CSNPs decreased as a result of the reaction of *A. vera* gel content with CSNPs or may be due to sonication before TEM examination. According to a recent study performed, CSNPs particles had a size ranging from 15 nm to 150 nm [41]. Spherical CSNPs with a size from 20 to 100 nm were visualized via TEM [39].

FTIR spectroscopy was performed to characterize the chemical structure of CSNPs, *A. vera* gel, and *A. vera* gel incorporated with CSNPs (Figure 5, Figure 6 and Figure 7). In a narrow range, peaks were observed at 3417.80, 3496.42, and 3442.24 cm^−1^ for *A. vera* gel, CSNPs, and *A. vera* gel incorporated with CSNPs, respectively, where these peaks are attributed to stretching vibrations of –NH_2_ and –OH groups.

A characteristic band of *A. vera* gel at 2922.94 cm^−1^ is associated with C–H or -CH_3_, these peaks are not detected in CSNPs alone or in *A. vera* gel incorporated with CSNPs, and they may be due to the reaction between chemical groups of *A. vera* gel with the chemical groups of CSNPs resulting from the appearance of new groups or the disappearance of detected groups.

The band at 1644.19 cm^−1^ indicates the presence of N-acetyl groups or the C=O stretching of amide I in CSNPs. In *A. vera* gel incorporated with CSNPs, the band at 1635.05 cm^−1^ that appeared may be due to the NH primary amine, while the band at 1731.22 cm^−1^ appearing in *A. vera* gel alone may be due to the C=O stretching vibration associated with acids, ketones, and aldehydes. The same approximate bands 1415.20 cm^−1^ and 1415.05 cm^−1^ appeared in CSNPs and *A. vera* gel incorporated with CSNPs, respectively, that may confirm the CH_2_ bending. In addition, peaks at 1153.61 and 1154.86 were observed for CSNPs and *A. vera* gel incorporated with CSNPs, respectively, that may be associated with the C–OH bending vibration and may detect the percentage of the crystalline phase. Characteristic bands in the range of 1383.70–1318.53 cm^−1^ are attributed to C–N stretching (amide III) in *A. vera* gel; however, Branca et al. [42] reported the presence of the group amide III at 1317 cm^−1^ CSNPs. The peak at 1252.83 cm^-1^ is characteristic of R=C-O-C and belongs to ethers in *A. vera* gel. The characteristic of the pyranoside ring absorption peak at 868.43 cm−1 (C-H ring vibration) was recognized in *A. vera* gel (Figure 5) as already identified by Nejatzadeh-Barandozi and Enferadi [43]. 

The appearance of a high-intensity band at 1018.99 cm−1 in CSNPs (Figure 6) may be attributed to the C−O and C−OH bonds in polysaccharides, as reported in a previous study [44] (Torres-Giner et al., 2017). The intense band at 1041.08 cm^−1^ is assigned to the existence of C–O and C–OH bonds in glucan units of *A. vera* after being incorporated with CSNPs (Figure 7). It can be presumed that this type of bond is designed among the various molecules (polysaccharides of *A. vera* gel and chitosan) interactions. CSNPs characterization with FTIR gave results parallel to those obtained in the reports conducted earlier [1,27].

### 3.3. Biological Activities of A. vera Gel and A. vera Gel Incorporated to CSNPs

#### 3.3.1. Anti-*Helicobacter pylori*

The current study demonstrates that the *A. vera* gel exhibited a moderate antibacterial potential against both resistant and susceptible *H. pylori* strains with inhibition zones of 16 and 16.5 mm, while CSNPs increased the antibacterial activity of *A. vera* gel to 28 and 30 mm, compared with using standard antibiotics that caused 20 and 25 mm inhibition zones, respectively (Table 2 and Figure 8). The inhibitor potential of *A. vera* gel was reported against *H. pylori* that could be attributed to the existence of anthraquinones [8]. In addition to the inhibition of *H. pylori*, Nostro et al. [9] mentioned that *A. vera* gel may prevent the *H. pylori* adhesion on gastric cells. A previous study on rats infected by *H. pylori* revealed that *A. vera* gel reduced the inflammation of gastric mucosa caused by *H. pylori* [45]. As reported in another study, not only *H. pylori* but other bacteria and fungi were inhibited by *A. vera* gel such as *Escherchia coli*, *Salmonella typhi*, *Bacillus subtilis*, *Staphylococcus aureus* [46], *Candida paraprilosis*, *Candida krusei* [5], and *C. albicans* [6]. MIC (15.62 and 3.9 µg/mL) and MBC (15.60 and 7.8 µg/mL) were assayed for both *A. vera* gel and *A. vera* gel incorporated with CSNPs, respectively, against a resistant strain of *H. pylori* (Table 3). The MBC/MIC Index was 1 and 2 for both *A. vera* gel and *A. vera* gel incorporated with CSNPs, respectively, indicating its bactericidal properties, while an MBC/MIC index of the current samples more than 4 revealed their bacteriostatic activity. These findings may well have an impact on the antimicrobial resistance phenomenon of *H. pylori*, proposing the *A. vera* inner gel as a promising effective natural agent with CSNPs for the treatment of *H. pylori* infection. Antibacterial properties of *A. vera* inner gel were demonstrated against both susceptible and resistant *H. pylori* strains [8]. These results may resolve the multi-drug-resistance phenomenon particularly in cases of *H. pylori*. Furthermore, as proposed by Pandey and Mishra [47], the inner gel of *A. vera* could be greatly effective when taken orally, because, inside a living human as well as an animal, both anthraquinones and acemannans as phytoconstituents of *A. vera* gel were able to guarantee its complete activity.

#### 3.3.2. Antioxidant Activity

A vital role of antioxidants was reported regarding the prevention of numerous diseases associated with oxidative stress; therefore, the antioxidant activity of *A. vera* gel alone and incorporated with CSNPs was determined. Data in the table showed that the antioxidant activity increased with the concentration increment in a dependent manner. As noticed, *A. vera* gel incorporated with CSNPs encouraged the antioxidant potential compared with *A. vera* gel alone particularly at high concentrations (Table 4 and Figure 9). DPPH Scavenging (%) was 68.1 and 75.2% at 1000 μg/mL using *A. vera* gel extract and *A. vera* gel incorporated with CSNPs, respectively. IC_50_ was 138.82 μg/mL using *A. vera* gel extract, while it became 81.7 μg/mL incorporated with CSNPs. In the current study, we identified and quantified 13 phenolic and flavonoid compounds (Table 3), confirming that the *A. vera* gel is a rich source of well-known antioxidant constituents [35]. The previous literature indicated that there is a correlation among the antioxidant properties and the content of phenolic compounds [48]. Therefore, the antioxidant potential of *A.vera* can be influenced according to development stages that are associated with changes in active compounds ingredients as well, as reported previously [46]. Kesharwani et al. [29] reported the enhancement of the *A. vera* gel antioxidant and good stability properties when loaded with CSNPs.

#### 3.3.3. Anti-Hemolytic Activity 

The anti-hemolytic activity increased as a result of *A. vera* gel incorporated with CSNPs compared with using *A. vera* gel alone (Figure 10), where the hemolysis inhibition was 40.0 and 75.4%, while it was 29.2 and 63.9% at 100 and 1000 mg/mL, respectively.

Generally, the hemolysis inhibition increased with the concentration of *A. vera* gel alone or *A. vera* gel incorporated with CSNPs in a dose-dependent mode. The results were compared with the hemolysis inhibition caused by the standard indomethacin where lysis of erythrocytes was shown to decrease with the increase in concentration. Sharifi-Rad et al. [16] according to a literature review concluded that pre-clinical and clinical investigations promote the application of CSNPs in nanomedicine. The effects of *A. vera* with CSNPs on inflammation and wound healing were studied [4], where *A. vera* gel was effective and becoming more effective when incorporated with CSNPs.

### 3.4. Molecular Docking Study 

Molecular docking is a method in molecular modeling that predicts the preferred orientation of one molecule to another when they are bonded in order to create a stable complex. Using scoring functions, the preferred orientation is utilized to determine the strength of connection or binding affinity between two molecules. 

The goal of molecular docking is to create an optimum conformation for both the protein and the ligand, as well as a relative orientation between the two, in order to reduce the total system’s free energy. Molecular recognition is important for enhancing basic biomolecular interactions including enzyme–substrate, drug–protein, and drug–nucleic acid interactions.

Here, we study the interaction between chlorogenic acid, chitosan, and pyrocatechol with the crystal structure of the *H. pylori* (4HI0) protein. The interaction among protein and compounds was visualized (Figure 11). 

The representative key for the types of interaction between chitosan, chlorogenic acid and pyrocatechol with 4HI0 protein of *H. pylori* was illustrated in Figure 12. 

The best fitted poses adopted by the compounds, binding energy scores, and energies are presented in Table 5. Chlorogenic acid exhibited a more potent binding energy than chitosan and pyrocatechol, which was observed as −6.4876 kcal/mol. In a recent study, chlorogenic acid had a strong interaction with the active site bound to chain (A) of the *E.coli* 7C7N protein, and its energy value was −6.0422 kcal mol^-^ [49]. In addition, a molecular docking report indicated the appearance of a superior negative score of free binding energy as a result of the application of chlorogenic acid on *Proteus vulgaris* and Human coronavirus (HCoV 229E), validating its application for inhibiting the bacterial and viral propagation [50]. A list of hydrogen bonds between all compounds with the examined protein is presented in Table 6.

Chlorogenic acid interacted via the (4HI0) protein with O 17 and O 23 by donating or accepting their H atoms through O ALA 41 and NZ LYS 195 of receptors.The interaction between chitosan and the active site bond of (4HI0) revealed the presence of a hydrogen donor atom between N 15 and N 67 in the ligand and the O ALA 41 and OD1 ASP 40 amino acid residue, in addition to the noncovalent molecular interaction (Cation-Pi) between the N 21 atom in the ligand and 6-ring TYR 48 amino acid receptor.The docked pyrocatechol with receptor active sites of (4HI0) indicated the presence of a hydrogen acceptor atom between O 41 of the ligand and the NZ LYS 195 amino acid residue with a distance of 2.85 ^o^A.

The docking pose and types of interaction agreed with the experimental results of the antibacterial activity of the main constituents of *A. vera* gel and CSNPs against *H. pylori*. 

## 4. Conclusions 

From HPLC analysis, *A. vera* gel is rich with various powerful antimicrobial and antioxidant constituents comprising flavonoids and phenolic contents. CSNPs displayed a relevant enhancement of the bacteriostatic activity of *A. vera* gel against *H. pylori*, as well as antioxidant and hemolysis inhibition. The experimental findings of the efficacy of *A. vera* gel and *A. vera* gel incorporated with CSNPs against *H. pylori* were documented. A docking study on the interaction of chlorogenic acid and pyrocatechol as the major constituent of *A. vera* gel, as well as CSNPs with a 4HI0 protein of *H. pylori*, confirmed the anti- *H. pylori* activity.

## Figures and Tables

**Figure 1 polymers-14-02994-f001:**
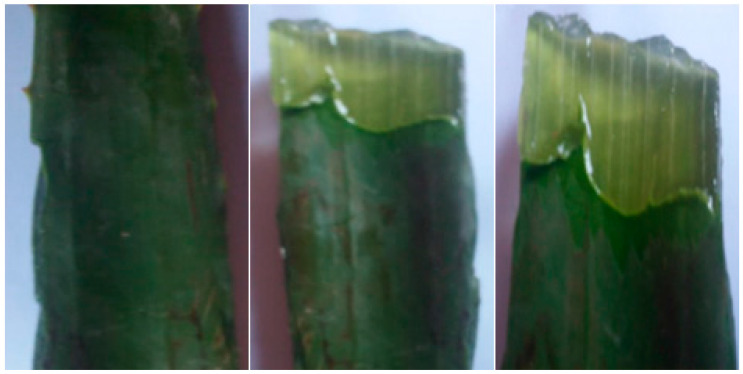
*A. vera* leaf showing outer green rind and viscous clear liquid (gel).

**Figure 2 polymers-14-02994-f002:**
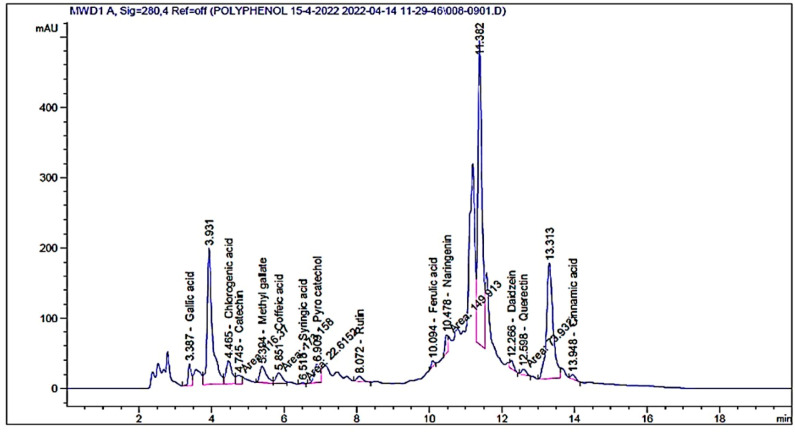
HPLC chromatograms of detected flavonoids and phenolic acids content of *A. vera* gel extract.

**Figure 3 polymers-14-02994-f003:**
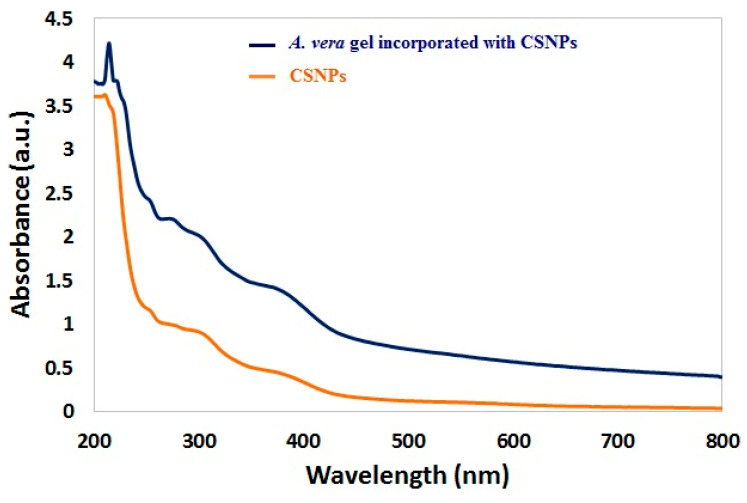
UV–vis spectra of CSNPs and *A. vera* gel incorporated with CSNPs.

**Figure 4 polymers-14-02994-f004:**
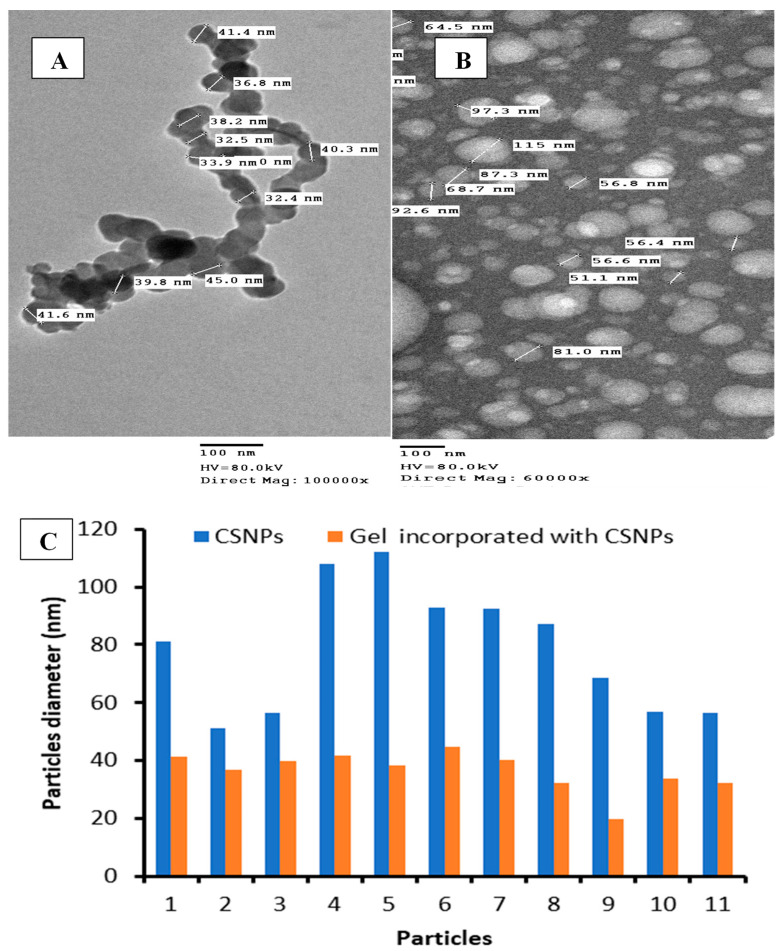
TEM of synthesized *A. vera* gel incorporated with CSNPs (**A**), CSNPs (**B**), and diameter of some particles (**C**).

**Figure 5 polymers-14-02994-f005:**
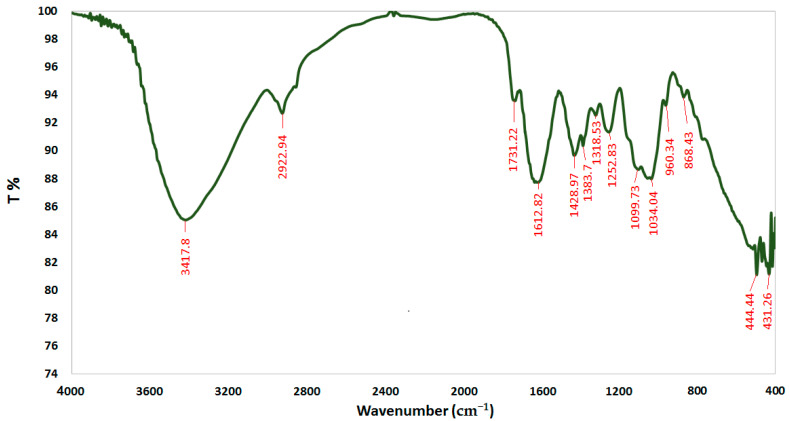
FTIR spectra of *A. vera* gel.

**Figure 6 polymers-14-02994-f006:**
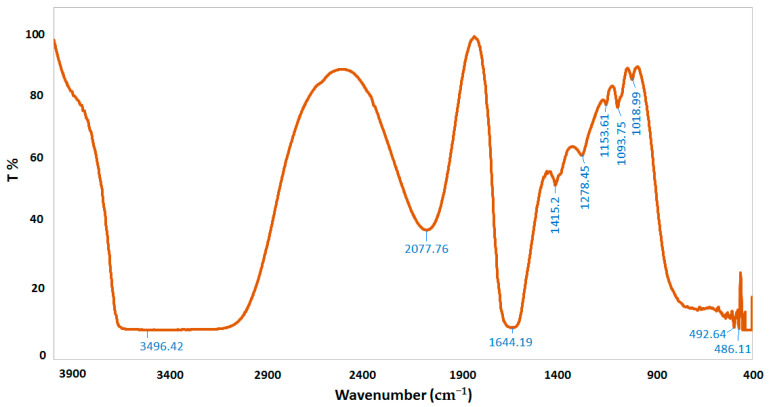
FTIR spectra of CSNPs.

**Figure 7 polymers-14-02994-f007:**
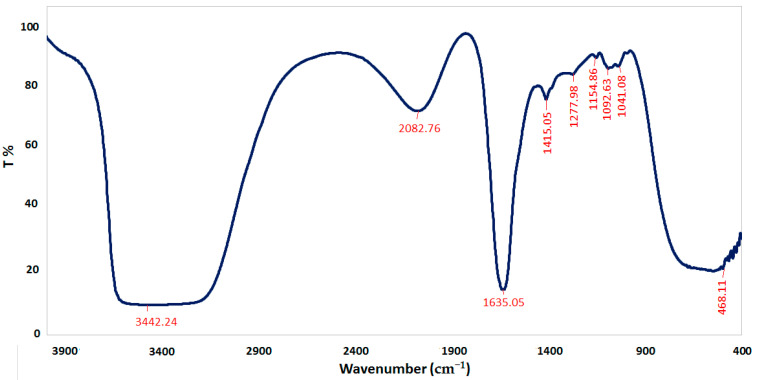
FTIR spectra of *A. vera* gel incorporated to CSNPs.

**Figure 8 polymers-14-02994-f008:**
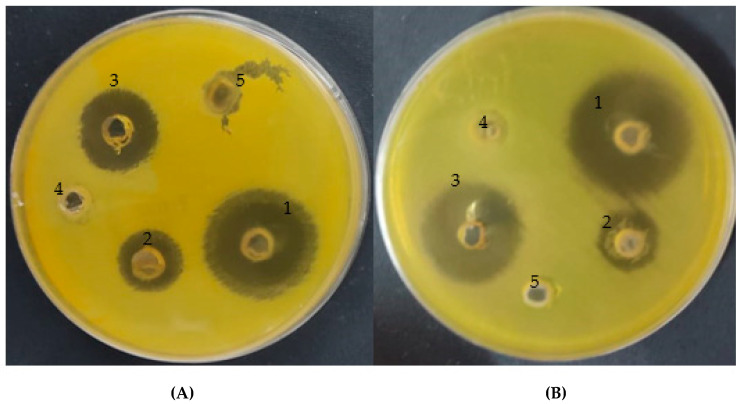
Inhibitory action of *A. vera* gel incorporated with CSNPs (1), *A. vera* gel (2), positive control (3), CSNPs (4), and acetic acid (5) against resistant strain (**A**) and sensitive strain (**B**) of *Helicobacter pylori*.

**Figure 9 polymers-14-02994-f009:**
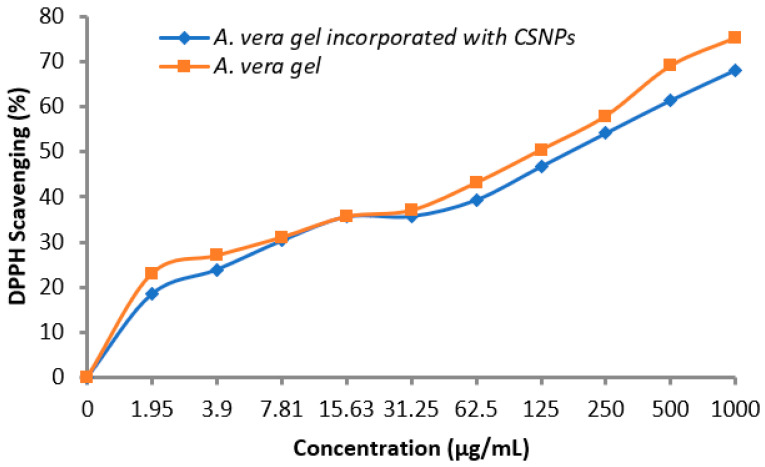
Antioxidant activity of *A. vera* gel and *A. vera* gel incorporated with CSNPs.

**Figure 10 polymers-14-02994-f010:**
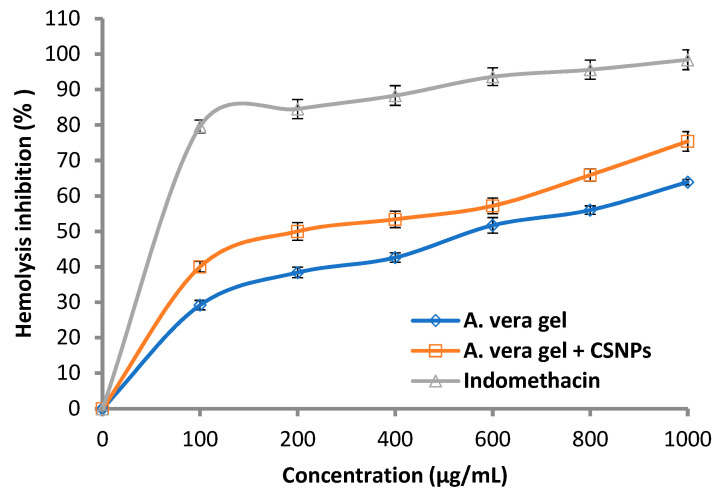
Hemolysis inhibition (%) of *A. vera* gel, *A. vera* gel incorporated with CSNPs, and indomethacin.

**Figure 11 polymers-14-02994-f011:**
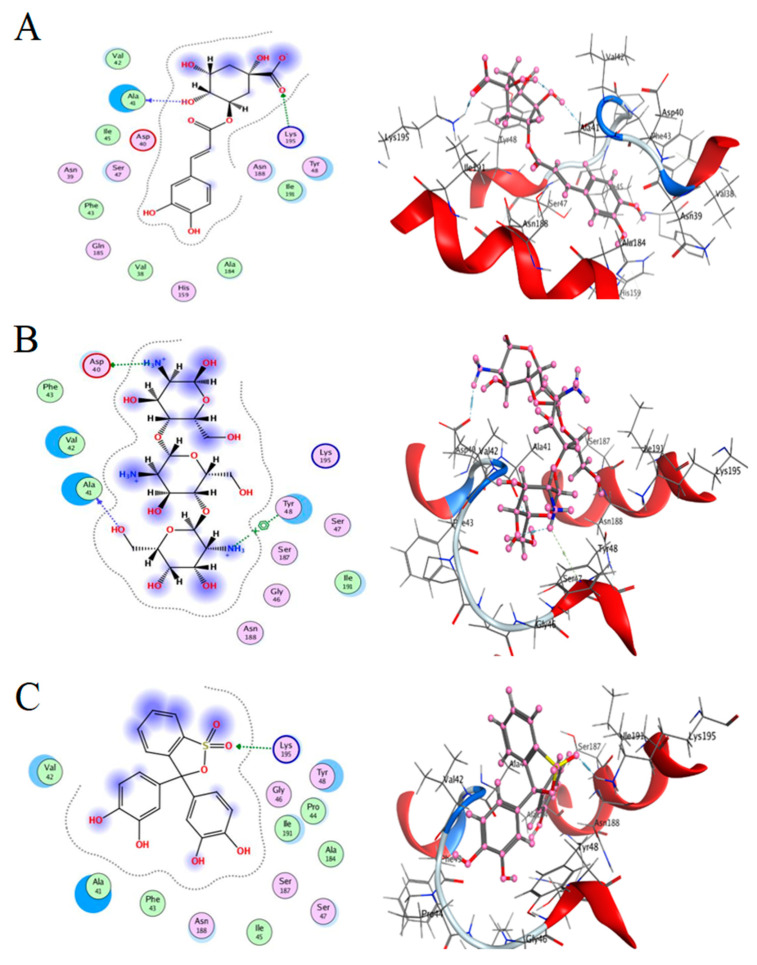
Docking interactions of certain compounds of *A. vera* gel extract.2D and 3D diagrams show the interaction between chlorogenic acid and active sites of 4HI0 protein (**A**); 2D and 3D diagrams show the interaction between chitosan and active sites of 4HI0 protein (**B**); 2D and 3D diagrams show the interaction between pyrocatechol and active sites of 4HI0 protein (**C**).

**Figure 12 polymers-14-02994-f012:**
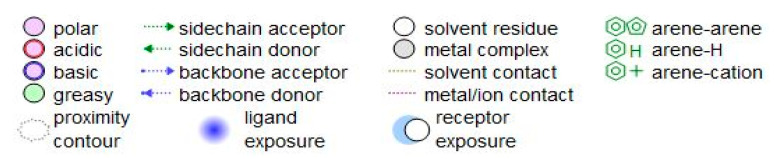
The representative key for the types of interaction between chitosan and pyrocatechol.

**Table 1 polymers-14-02994-t001:** Flavonoids and Phenolic acids content of *M. pulegium* extract identified by HPLC.

Flavonoids			Phenolic Acids		
RT *	Compound	Concentration (µg/mL)	RT*	Compound	Concentration (µg/mL)
3.387	Gallic acid	466.99	8.072	Rutin	194.39
4.465	Chlorogenic acid	1637.09	10.094	Ferulic acid	105.24
4.745	Catechin	1100.43	10.478	Naringenin	528.78
5.394	Methylgallate	663.41	12.266	Daidzein	190.72
5.851	Caffeic acid	494.05	12.598	Quercetin	295.25
6.518	Syringic acid	71.98	13.948	Cinnamic acid	37.50
6.909	Pyrocatechol	1552.92			

**RT ***, retention time.

**Table 2 polymers-14-02994-t002:** *Anti-Helicobacter pylori* using *A. vera* gel and *A. vera* gel incorporated with CSNPs.

Strain Test	Inhibition Zone (mm)				
	*A. vera* gel INCORPORATED with CSNPs	*A. vera* Gel	Control	CSNPs	Acetic Acid
** *H. pylori (1)* **	28	16	20	0	0
** *H. pylori (2)* **	30	16.5	25	0	0

**Table 3 polymers-14-02994-t003:** MIC and MBC of *A. vera* gel and *A. vera* gel incorporated with CSNPs against resistance strain of *H. pylori*.

Compounds	MIC (µg/mL)	MBC µg/mL)	MBC/MIC Index
*A. vera* gel incorporated with CSNPs	3.9	7.8	2.0
*A. vera* gel	15.62	15.60	1.0

**Table 4 polymers-14-02994-t004:** Antioxidant activity of *A. vera* gel and *A. vera* gel incorporated with CSNPs.

Concentration (μg/mL)	*A. vera* gel Incorporated with CSNPs				*A.vera* Gel			
	O.D Mean	DPPH Scavenging (%)	SD	SE	O.D Mean	DPPH Scavenging (%)	SD	SE
1000	0.380	75.2	0.014	0.004	0.497	68.1	0.002	0.001
500	0.482	69.0	0.003	0.001	0.606	61.4	0.012	0.004
250	0.665	57.9	0.007	0.002	0.727	54.1	0.012	0.004
125	0.787	50.4	0.004	0.001	0.848	46.7	0.003	0.001
62.5	0.907	43.1	0.014	0.004	0.969	39.3	0.006	0.002
31.25	1.007	37.0	0.002	0.001	1.028	35.7	0.003	0.001
15.63	1.029	35.6	0.002	0.001	1.029	35.6	0.002	0.001
7.81	1.105	31.0	0.003	0.001	1.117	30.3	0.003	0.001
3.9	1.170	27.0	0.006	0.002	1.221	23.9	0.014	0.005
1.95	1.237	23.0	0.004	0.001	1.311	18.5	0.005	0.002
0	1.641	0.00	0.010	0.003	1.641	0.0	0.010	0.003
IC_50_	81.7 μg/mL				138.82 μg/mL			

**Table 5 polymers-14-02994-t005:** The best possible conformations of compounds inside the protein central activity.

Comp	Mol	S	rmsd_ref-n	E_conf	E_place	E_score1	E_refine	E_score2
Chlorogenic acid	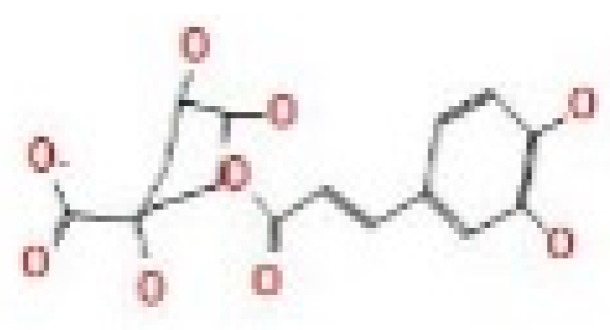	−6.4876	3.5275	−13.4731	−59.0633	−11.5790	−31.1223	−6.4876
Chlorogenic acid	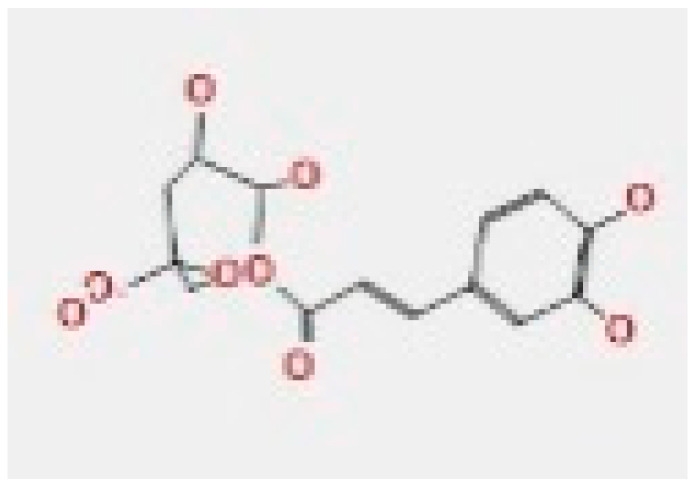	−6.4831	2.4795	−10.5458	−66.7119	−13.6501	−32.0878	−6.4831
Chlorogenic acid	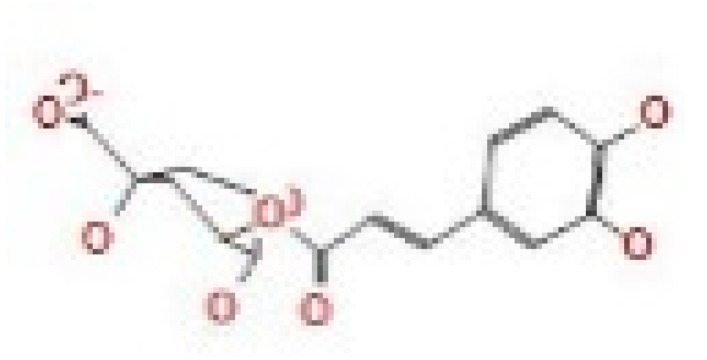	−6.2798	1.6063	−16.8045	−77.5656	−12.1824	−28.4793	−6.2798
Chlorogenic acid	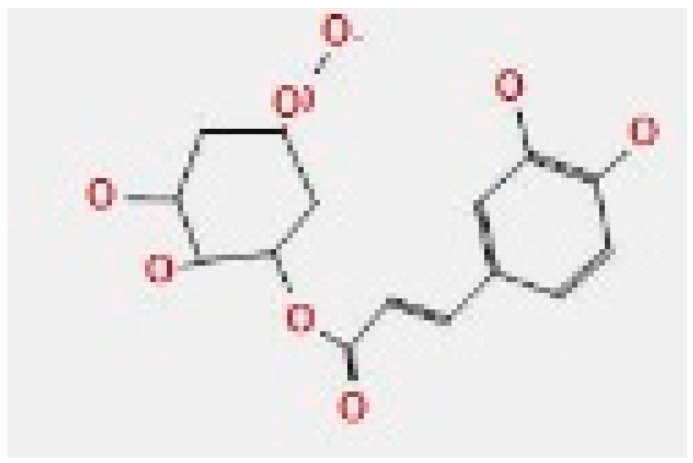	−6.2138	2.1646	−4.0346	−85.5775	−13.3926	−31.8557	−6.2138
Chlorogenic acid	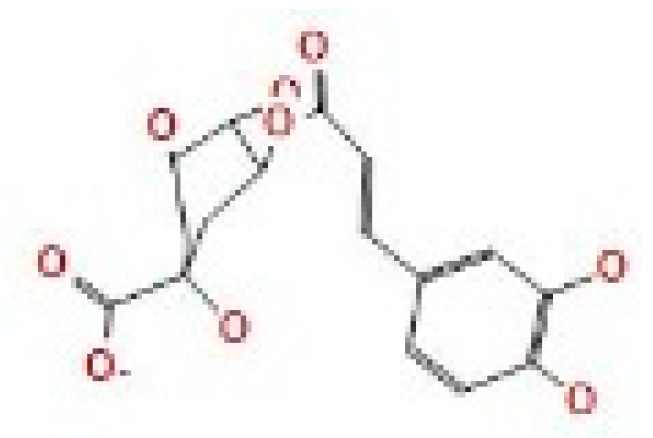	−6.0663	2.0747	−1.4078	−61.5997	−12.1744	−29.5917	−6.0663
Chitosan	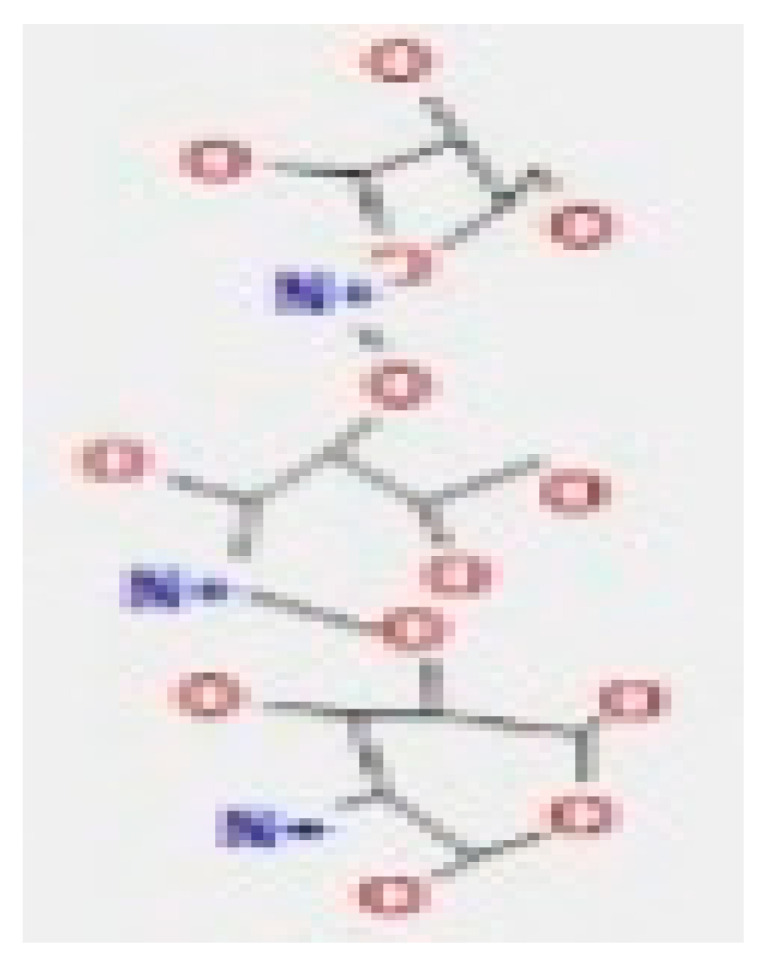	−5.8051	3.3384	121.2666	−112.7581	−13.5777	−17.1217	−5.8051
Chitosan	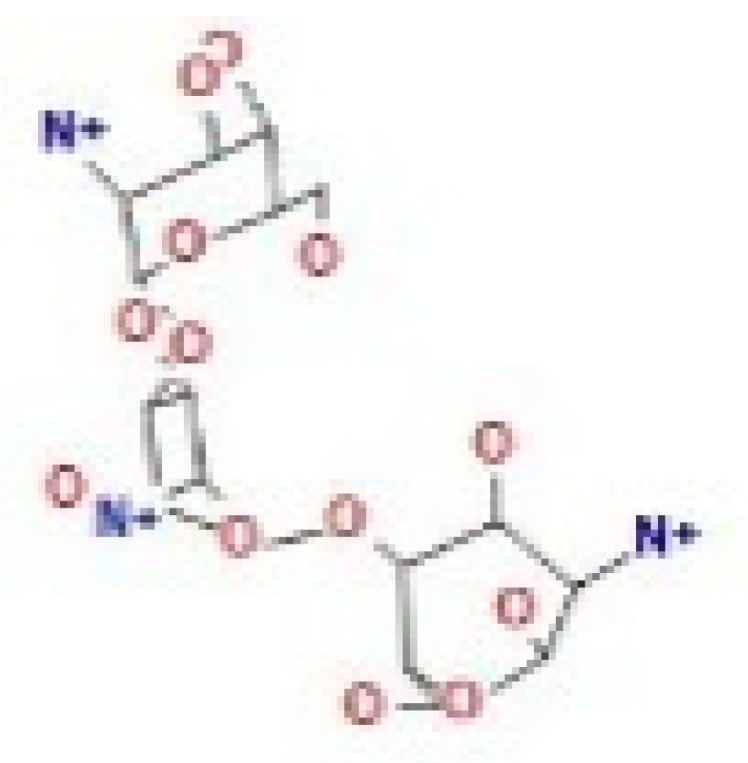	−5.3893	3.6858	112.1133	−92.1878	−12.0519	−30.4200	−5.3893
Chitosan	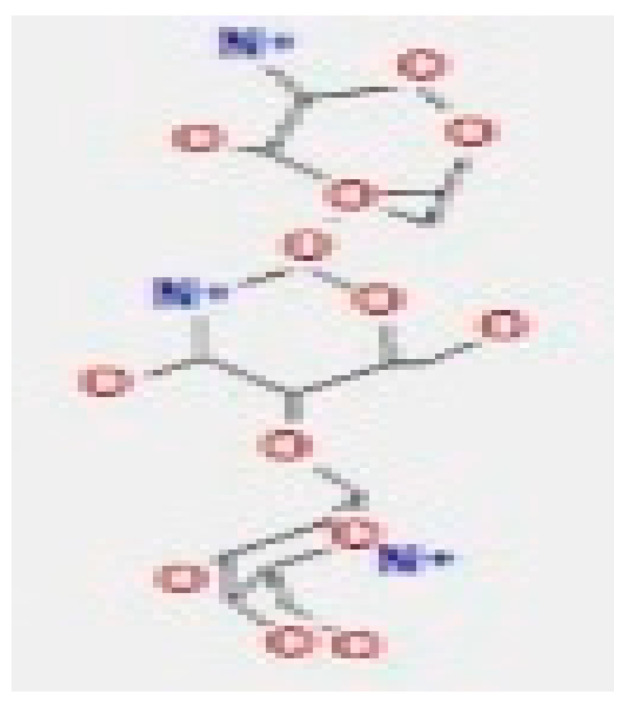	−5.3659	1.6801	93.4811	−83.7728	−8.3993	−22.3280	−5.3659
Chitosan	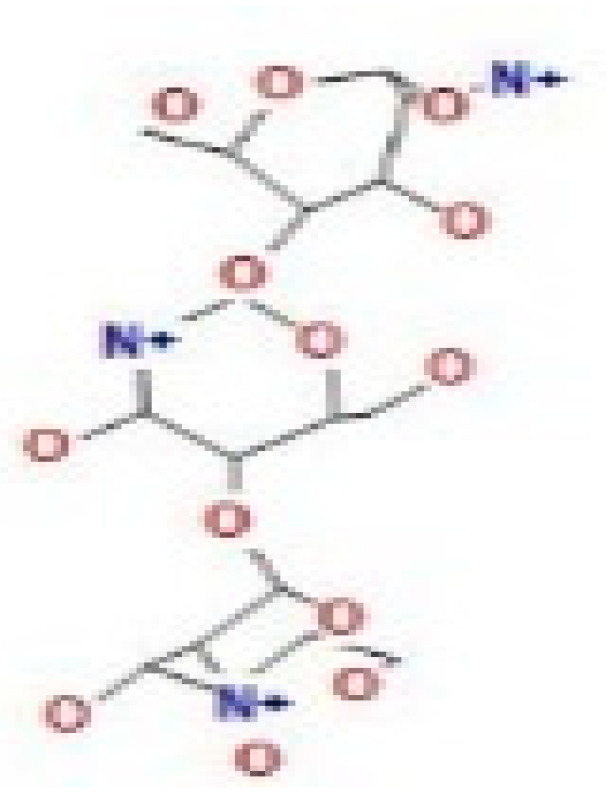	−5.1855	4.3113	99.2838	−79.9617	−8.8196	−21.6441	−5.1855
Chitosan	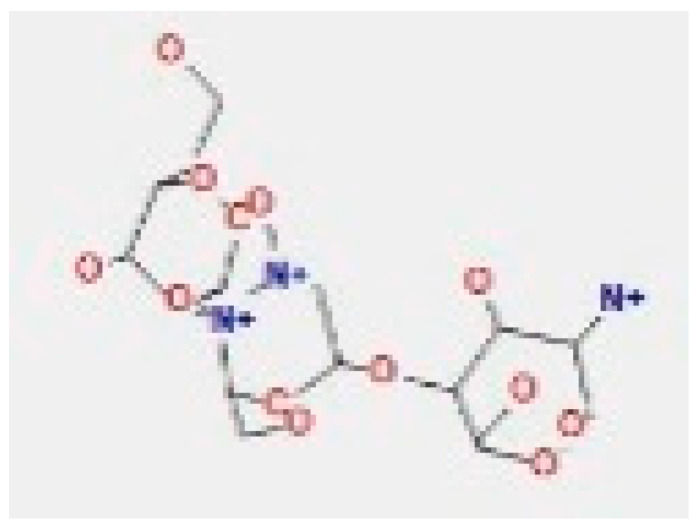	−5.1464	2.3545	97.8875	−28.5749	−9.2994	−26.0396	−5.1464
Pyrocatechol	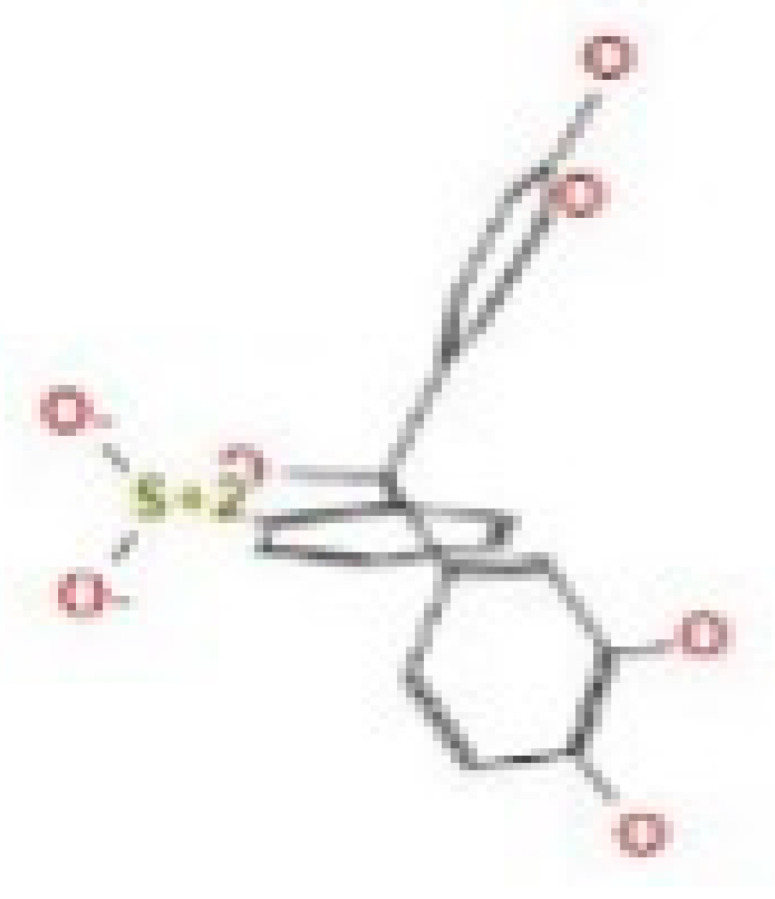	−5.4074	1.3093	32.0714	−36.7848	−9.6168	−22.1137	−5.4074
Pyrocatechol	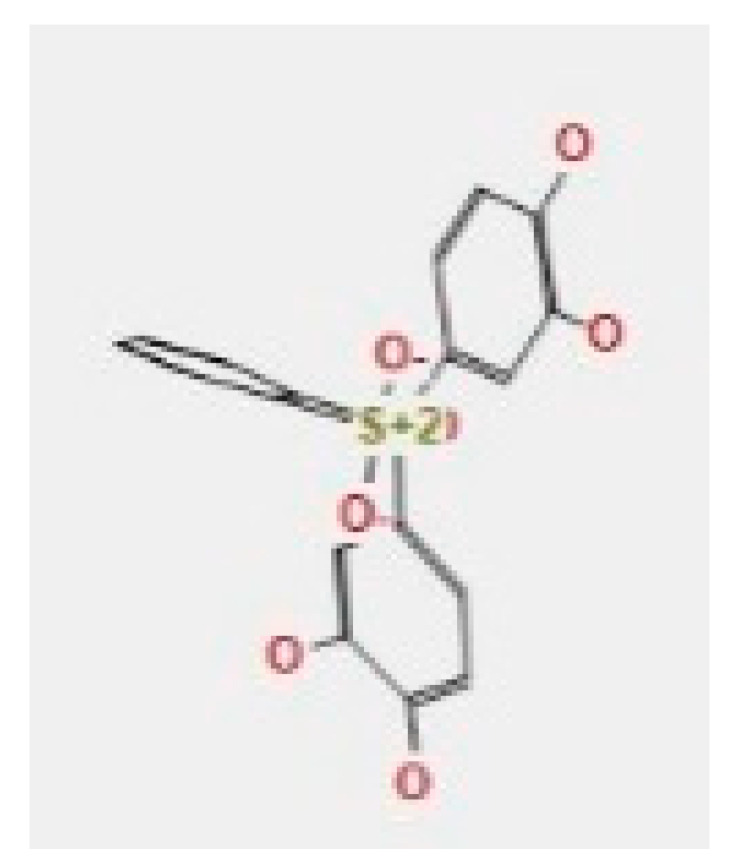	−5.1043	3.5432	22.2566	−51.3658	−9.6291	−26.8970	−5.1043
Pyrocatechol	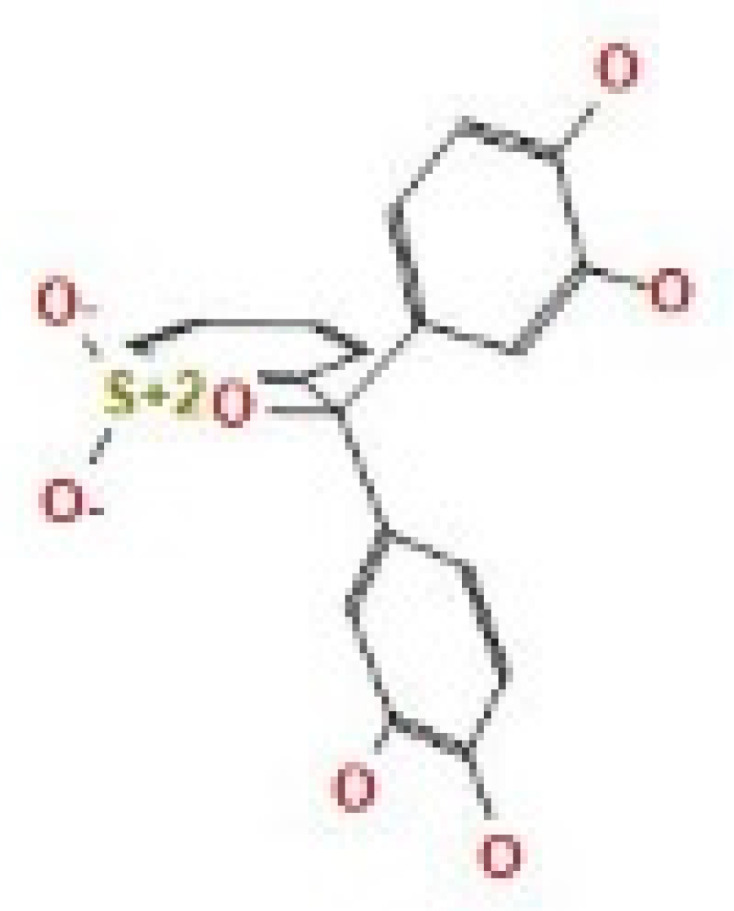	−4.9689	4.4384	25.2747	−55.7667	−10.2540	−25.9606	−4.9689
Pyrocatechol	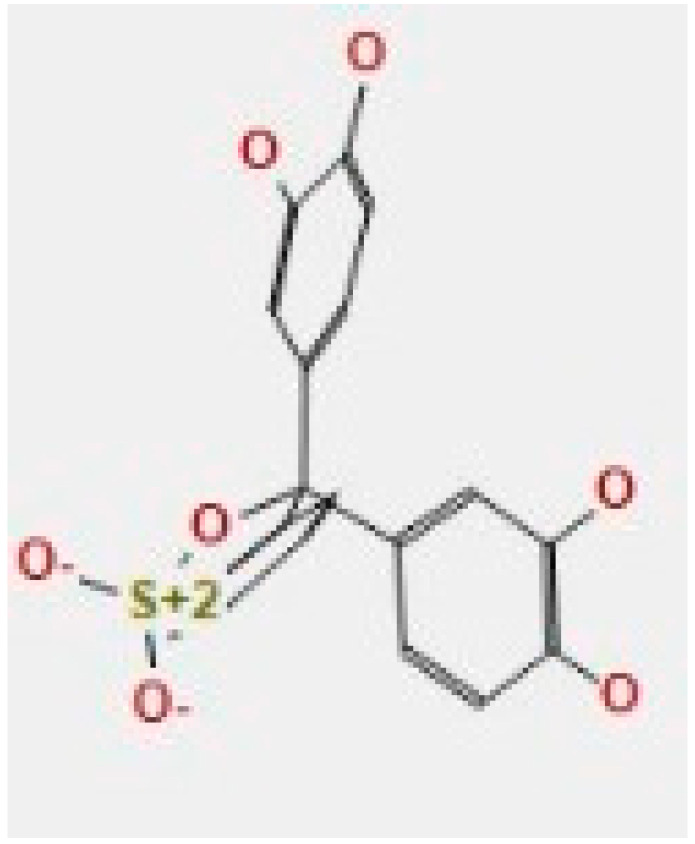	−4.9671	1.2911	48.9902	−62.7305	−9.7276	−14.6363	−4.9671
Pyrocatechol	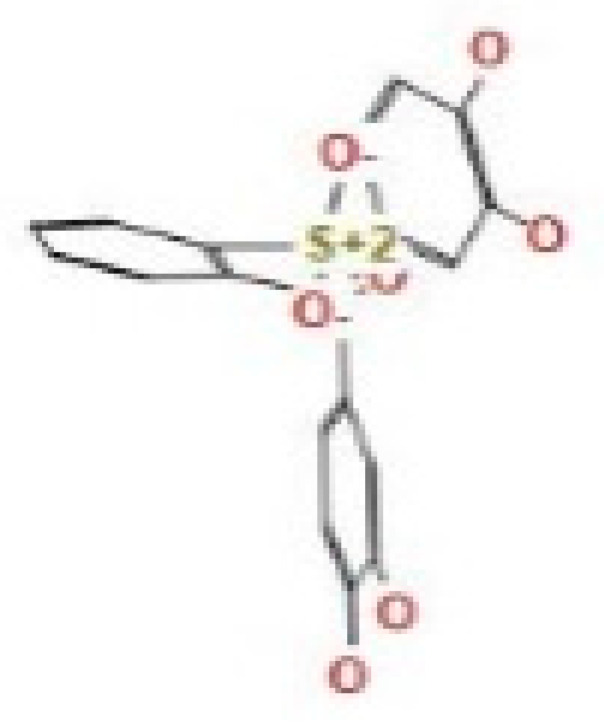	−4.9025	2.3984	25.1460	−73.3035	−9.6229	−24.9072	−4.9025

S, final score (which is the score of the last stage that was not set to none); rmsd, the root-mean-square deviation of the pose, in Å, from the original ligand; rmsd_refine, the root-mean-square deviation between the pose before refinement and the pose after refinement; E_conf, the energy of the conformer; E_place, score from the place-ment stage; E_score 1 and E_score 2, score from rescoring stages 1 and 2, respectively; E_refine, score from the refinement stage, calculated to be the sum of the van der Waals electrostatics and solvation energies, under the Generalized Born solvation model (GB/VI).

**Table 6 polymers-14-02994-t006:** Interaction of chlorogenic acid, pyrocatechol, and chitosan with 4HI0 Protein.

Chlorogenic Acid	Receptor	Interaction	Distance	E (kcal/mol)
O 17	O ALA 41 (A)	H- donor	2.98	−0.8
O 23	NZ LYS 195 (A)	H- acceptor	2.96	−3.3
Pyrocatechol	Receptor	Interaction	Distance	E (kcal/mol)
O 41	NZ LYS 195 (A)	H- acceptor	2.85	−6.8
Chitosan	Receptor	Interaction	Distance	E (kcal/mol)
O 15	O ALA 41 (A)	H- donor	2.96	−2.1
N 67	OD1 ASP 40 (A)	H- donor	2.93	−15.4
N 21	6-ring TYR 48 (A)	Cation-Pi	3.93	−1.4

## Data Availability

All data that support the findings of this study are available within the article.

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
