# Peer review of "Molecular Docking and Efficacy of Aloe vera Gel Based on Chitosan Nanoparticles against Helicobacter pylori and Its Antioxidant and Anti-Inflammatory Activities"

_polymers, 2022, doi:10.3390/polym14152994_

Round 1

Reviewer 1 Report

Dear authors,

This paper deals with the molecular Docking and Efficacy of Aloe vera gel Based on chitosan nanoparticles Against Helicobacter pylori. This is an interesting paper but several issues needs to be attended to before being considered for publication.

Introduction section.

Line 65...plant and pathogens names are in higher letter size, check this along with the manuscript.

Line 72 and 73, check and correct the name of glucosamine and N-acetylglucosamine.

A lot of work combining chitosan nanoparticles and Aloe vera has been published, please discuss their findings in the introduction section. Highlight the main difference with this work. 

Methodology section.

Which polyphenols and flavonoids were determined by HPLC. Who supplied them?

Line 120...multi-wavelength detector...what does it means? UV- detector? 

Describe in separate methodologies, the SUV and TEM and FTIR. In latter case (FTIR) how many scans and resolution per sample did you used?) Sample was measured in FTIR or ATR mode?

Line 35...how many replicates were used?.

Results section.

Line 293 1nd 294.. At which group corresponds the 1731 band from aloe vera gel?. Which other characteristic bands appear. An improved discussion of FTIR analysis must be presented.

The authors must Improve the discussion with other papers in the whole manuscript.

Author Response

Line 65...plant and pathogens names are in higher letter size, check this along with the manuscript.

its check in all  manuscript

Line 72 and 73, check and correct the name of glucosamine and N-acetylglucosamine.

its corrected to β-(1-4)-linked d-glucosamine and N-acetyl-d-glucosamine 

A lot of work combining chitosan nanoparticles and Aloe vera has been published, please discuss their findings in the introduction section. Highlight the main difference with this work. 

Phytoconstituents and its concentrations of A. vera as well as other plants may differ according to cultivation soil, climatic changes, type of fertilizers and extraction methods. In addition,  most of studies on the incorporation  of A. vera gel with CSNPs focused on wound healing and antimicrobial activity against certain microorganisms but not against H. pylori. Therefore, the aim of this study was to enhancement  the biological activities of A. vera gel with the use of CSNPs as a natural safe compound. We studied the effects of this incorporation on H. pylori growth, antioxidant activity and anti-inflammatory. Moreover, the  docking  study of the main components of A. vera gel on  H. pylori.

Methodology section.

Which polyphenols and flavonoids were determined by HPLC. Who supplied them?

Line 120...multi-wavelength detector...what does it means? UV- detector? 

UV- detector

Describe in separate methodologies, the SUV and TEM and FTIR. In latter case (FTIR) how many scans and resolution per sample did you used?) Sample was measured in FTIR or ATR mode?

OK its separated

Line 35...how many replicates were used?.

three replicates

Results section.

Line 293 1nd 294.. At which group corresponds the 1731 band from aloe vera gel?. Which other characteristic bands appear. An improved discussion of FTIR analysis must be presented.

band at 1731.22 cm−1 appeared in A. vera gel alone that may be due to C=O stretching vibration associated to acids, ketons and aldehydes . 

Reviewer 2 Report

Review Manuscript Number: polymers-1765013

Title: Molecular Docking and Efficacy of Aloe vera gel Based on chitosan nanoparticles Against Helicobacter pylori

The present study focuses on combining chitosan nanoparticles and aloe vera gel to treat H. Pylori

The authors mechanically extracted the aloe vera gel from the plants and then concentrated the gel by evaporation at 50°C and vacuum (which has not been properly informed). The main components in the extracts were characterized by HPLC being Chlorogenic acid, Catechin, Pyrocatechol, Methylgallate, Naringenin, Caffeic acid, and Gallic acid the most abundant ones. 

The characterization of the obtained materials by UV-Vis and FTIR is not conclusive but could indicate the combination of the gel with the CSNPs. However, the inhibition activity of the combined material is superior to the one of each compound by itself. The authors provided the antioxidant activity of each material as a possible explanation of the enhanced inhibitory property of CSNPs - A. vera gel, reporting higher DPPH scavenging for the combined material than that of A vera gel. Also, the hemolysis inhibition was higher for the combined material than the gel alone, proving that a synergistic effect exists when the CSNPs are used with the Aloe vera gel extract. 

The authors also included the molecular modeling for the docking of the main constituents of material (namely, chlorogenic acid, chitosan, and pyrocatechol) with a crystalline structure of the H pylori (4HIO) protein. 

Results showed that chlorogenic acid exhibited a better binding with the protein than the other two constituents. A better interpretation and description of these results should be included in the manuscript. In the current version, there is no connectivity between these results and the other ones in the manuscript. 

Minor comments

- Abstract

There are places with no spaces between words or between words and numbers that should be corrected. For example: line 19, line 20, 

Consider using “A. Vera with CSNPs” or “A. Vera loaded with CSNPs” instead of “A. Vera incorporated with CSNPs”. 

Line 19-20 - I believe the authors wanted to express something like: “CSNPs and A. Vera gel incorporating CSNPs were examined via TEM indicating mean sizes of 83.46 nm and 36.54 nm, respectively”. Or something like that. 

Line 22 - Please correct to “…using A. vera gel with inhibition zones of 16 and 16.5 nm…”

Line 23 – please correct to “…while A. vera gel with CSNPs exhibited the highest inhibition zones of 28 and 30 nm with resistant and sensitive strains, respectively.”

MIC and MBC are not usual abbreviations for readers of Polymers journal. Please consider describing their meaning in the abstract “minimal inhibitory concentration (MIC) and “minimal bactericidal concentration (MBC)”

A. vera should be used in italics every time. Please revise the whole manuscript, and correct that. e.g. Line 27, 28, 29

Line 28 - There is a double parenthesis in “… inhibition (( %compared …” and no closing parenthesis at the end of the sentence. 

The abstract is poorly written, and there are too many grammar, language, and formatting issues. With great effort, it becomes possible to understand the information provided by the authors. So, please revise the whole manuscript, aiming for a clear and straightforward structure with an appropriate writing quality for a scientific manuscript.

I strongly suggest the authors to avoid submitting manuscripts with this writing quality, since it sets a bad precedent for them.

- Keywords

Typically, keywords are used to expand search results from the words already included in the title, so there is no sense in including the same words already contained in the title. Consider using different terms since the current title already have all the keywords . 

- Introduction

There are several double spaces within the text, please revise the whole manuscript and correct them. e.g. Lines 37, 45, 47, 49, 56, 59, 62, 80 (x3), 81, 82, etc.

Line 38 – correct to “… succulent plant that belongs to the liliaceous …”

Line 39 – correct to “…plant, which contains …”

Line 46 – please correct the sentence to “The anti-oxidative activity reduces …”

Line 47 – Something is missing in the sentence “The viscosity of A. vera gel due to its content of the sugar glucomannan”: Please complete the sentence.

Line 52. Correct to “… the stomach of humans…”

Line 57 – Correct to “… problem for public health, and it has been of interest …”

Line 59-60 – Correct to “… of a novel strategy combining nanoparticles and natural compounds becomes a medical …”

Line 68 – I believe the references should be placed before “From the applied…”

Line 70 – Something is missing in the sentence “…, chitosan nanoparticles (CSNPs) that utilize lately on large scale level in particularly in medicinal fields…” maybe the authors wanted to express something like “From the applied NPs, chitosan nanoparticles (CSNPs) have been lately used in large scales for medical applications…” if that is the idea of that sentence consider correcting it and providing more relevant information (which applications, or something about the large scale process).

Line 73 – Something is missing in “…1, 4-N-acetyl glucose amine. is a cationic polymer.”

Line 74 – Correct to “… in previous literature…”

Line 75 – Correct to “…, to inhibit the development of scar tissue and for protein delivery, as well as other biomedical applications…”

Line 83 – Correct to “…, the gel gains several…”

Line 88 – Correct to “… cell destruction. The activity of…”

Line 93-95 – Correct to “…). This study aimed to enhance the use of A. vera with CSNPs as a natural and safe compound. 

The proposed aim does not seem to fit the one from the investigation. The main goal in the manuscript seems to be the combination of A. vera gel with CSNPs for the specific application of treating H. pylori. In the current version, the proposed main goal is only the combination of the gel and the nanoparticles. Please revise the aims and objectives of the study and be clear on what is going to be answered within the results and conclusions. 

Since there are so many English, grammar, and formatting errors in the manuscript, from this point and on, only content issues will be addressed. Please revise, and correct the whole manuscript before resubmitting it. Please avoid submitting manuscripts with this poor writing quality in the future. 

- Materials and Methods

As far as I understand from the manuscript, two types of products were prepared. First, a suspension of CSNPs in water with acetic acid, glycerol, and Tween 20; Then, the same suspension with the addition of A. vera gel. I am not able to get from the text what is the ratio between A. vera gel and CSNPs. 

What is the final product, the aqueous suspension? How do you discriminate the amount of A. vera gel within the nanoparticles and the amount dissolved in the aqueous solution?

I believe the authors performed the characterization of the products by TEM and FTIR with dry products. If so, Do they expect to have the same distribution that the one in the aqueous suspension?

- Results and discussions

How do you explain the size differences of CSNPs with and without A vera gel? It seems counterintuitive that the particles decrease in size when the gel is incorporated. 

What is the purpose of the results presented in Figure 4C? If the authors wanted to show a size distribution, increase the number of particles in the analysis and order them in size to provide a proper distribution. Just showing random sizes does not provide useful information. 

Please provide an interpretation of why the characteristic peak of A. Vera gel (assigned to C-H and CH3 functional groups at 2922.94 cm-1) is not present in the CSNPs combined with gel.

What was the goal of the FTIR analysis and comparison between materials? There are no solid or consistent results among them that prove the incorporation of the gel in the combined NP-gel material. Also, the interpretation of the spectra should be improved. 

Please provide a detailed and proper explanation of the sentence in line 408: “The docking pose and types of interaction were agreed with the experimental results”, what results are you referring to, and why do they are confirmed or agree with the modeling analysis.

Conclusions

The main results of the study have been included in the conclusions. However, It is hard to review the quality of the conclusions because of the poor writing quality of the manuscript. Maybe including specific information within this section could improve the conclusions, e.g. the most relevant constituents responsible for the antioxidant characteristics of the gel; the quantitative difference in the bacteriostatic activity of the assayed materials against H. pylori; etc.

Author Response

There are places with no spaces between words or between words and numbers that should be corrected. For example: line 19, line 20, 

its corrected

Consider using “A. Vera with CSNPs” or “A. Vera loaded with CSNPs” instead of “A. Vera incorporated with CSNPs”. 

very thanks  it was taken in our consideration in future papers

Line 19-20 - I believe the authors wanted to express something like: “CSNPs and A. Vera gel incorporating CSNPs were examined via TEM indicating mean sizes of 83.46 nm and 36.54 nm, respectively”. Or something like that. 

its  corrected to : indicating mean sizes of 83.46 nm and 36.54 nm, respectively”

Line 22 - Please correct to “…using A. vera gel with inhibition zones of 16 and 16.5 nm…”

its corrected

Line 23 – please correct to “…while A. vera gel with CSNPs exhibited the highest inhibition zones of 28 and 30 nm with resistant and sensitive strains, respectively.”

its changed

MIC and MBC are not usual abbreviations for readers of Polymers journal. Please consider describing their meaning in the abstract “minimal inhibitory concentration (MIC) and “minimal bactericidal concentration (MBC)”

its corrected

  1. verashould be used in italics every time. Please revise the whole manuscript, and correct that. e.g. Line 27, 28, 29

its corrected

Line 28 - There is a double parenthesis in “… inhibition (( %compared …” and no closing parenthesis at the end of the sentence. 

its corrected

The abstract is poorly written, and there are too many grammar, language, and formatting issues. With great effort, it becomes possible to understand the information provided by the authors. So, please revise the whole manuscript, aiming for a clear and straightforward structure with an appropriate writing quality for a scientific manuscript.

its revised

- Keywords

Typically, keywords are used to expand search results from the words already included in the title, so there is no sense in including the same words already contained in the title. Consider using different terms since the current title already have all the keywords .

very thanks : its  changed to : Evaluation; in vitro; Aloe vera; chitosan nanoparticles; Helicobacter pylor; therapeutic effects

- Introduction

There are several double spaces within the text, please revise the whole manuscript and correct them. e.g. Lines 37, 45, 47, 49, 56, 59, 62, 80 (x3), 81, 82, etc.

its corrected

Line 38 – correct to “… succulent plant that belongs to the liliaceous …”

its corrected

Line 39 – correct to “…plant, which contains …”

its corrected

Line 46 – please correct the sentence to “The anti-oxidative activity reduces …”

its corrected

Line 47 – Something is missing in the sentence “The viscosity of A. vera gel due to its content of the sugar glucomannan”: Please complete the sentence.

its changed to : The viscosity of A. vera gel may due to its content of the sugar glucomannan, in addition to approximately 200 active components which were detected in A. vera gel among proteins and its monomers, lipids, vitamins and polysaccharides

Line 52. Correct to “… the stomach of humans…”

its corrected

Line 57 – Correct to “… problem for public health, and it has been of interest …”

its corrected

Line 59-60 – Correct to “… of a novel strategy combining nanoparticles and natural compounds becomes a medical …”

its corrected

Line 68 – I believe the references should be placed before “From the applied…”

Its transferred before “From the applied…”

Line 70 – Something is missing in the sentence “…, chitosan nanoparticles (CSNPs) that utilize lately on large scale level in particularly in medicinal fields…” maybe the authors wanted to express something like “From the applied NPs, chitosan nanoparticles (CSNPs) have been lately used in large scales for medical applications…” if that is the idea of that sentence consider correcting it and providing more relevant information (which applications, or something about the large scale process).

its corrected to :chitosan nanoparticles (CSNPs) have been lately used in large scales for medical applications such as drug carrier and cosmetics 

Line 73 – Something is missing in “…1, 4-N-acetyl glucose amine. is a cationic polymer.”

its corrected to :The chemical structure of chitosan (cationic polymer) is fabricated of monomers of β-(1-4)-linked d-glucosamine and N-acetyl-d-glucosamine 

Line 74 – Correct to “… in previous literature…”

its corrected

Line 75 – Correct to “…, to inhibit the development of scar tissue and for protein delivery, as well as other biomedical applications…”

its corrected to : to inhibit the development of scar tissue and for protein delivery, as well as other biomedical applications

Line 83 – Correct to “…, the gel gains several…”

its corrected

Line 88 – Correct to “… cell destruction. The activity of…”

its corrected

Line 93-95 – Correct to “…). This study aimed to enhance the use of A. vera with CSNPs as a natural and safe compound. 

its corrected to (some correction also according to  another prof.   1st reviewer):    Phytoconstituents and its concentrations of A. vera as well as other plants may differ according to cultivation soil, climatic changes, type of fertilizers and extraction methods. In addition,  most of studies on the incorporation  of A. vera gel with CSNPs focused on wound healing and antimicrobial activity against certain microorganisms but not against H. pylori. Therefore, the aim of this study was to enhance the biological activities of A. vera gel with CSNPs as a natural and safe compound. We also studied the effects of this incorporation on H. pylori growth, antioxidant activity and anti-inflammatory. Moreover, the docking study of the main components of A. vera gel on H. pylori.

- Materials and Methods

As far as I understand from the manuscript, two types of products were prepared. First, a suspension of CSNPs in water with acetic acid, glycerol, and Tween 20; Then, the same suspension with the addition of A. vera gel. I am not able to get from the text what is the ratio between A. vera gel and CSNPs. 

it 10%

What is the final product, the aqueous suspension? How do you discriminate the amount of A. vera gel within the nanoparticles and the amount dissolved in the aqueous solution?

it 10%

I believe the authors performed the characterization of the products by TEM and FTIR with dry products. If so, Do they expect to have the same distribution that the one in the aqueous suspension?

To perform TEM and FTIR analysis , the sample must be dried

- Results and discussions

How do you explain the size differences of CSNPs with and without A vera gel? It seems counterintuitive that the particles decrease in size when the gel is incorporated. 

 under TEM, the size  appeared reduced

What is the purpose of the results presented in Figure 4C? If the authors wanted to show a size distribution, increase the number of particles in the analysis and order them in size to provide a proper distribution. Just showing random sizes does not provide useful information. 

To show the diameters of some selected randomly CSNPs and CSNPs incorporated with gel were recorded.

Please provide an interpretation of why the characteristic peak of A. Vera gel (assigned to C-H and CH3 functional groups at 2922.94 cm-1) is not present in the CSNPs combined with gel.

May be due to the reaction between chemical groups of  A. vera gel with the chemical groups of CSNPs  resulting of appearance of new groups or disappearance of detected groups

What was the goal of the FTIR analysis and comparison between materials? There are no solid or consistent results among them that prove the incorporation of the gel in the combined NP-gel material. Also, the interpretation of the spectra should be improved. 

To detected the chemical groups that appeared or disappeared as a result of combination among A. vera gel and CSNPs ., and its improved

Please provide a detailed and proper explanation of the sentence in line 408: “The docking pose and types of interaction were agreed with the experimental results”, what results are you referring to, and why do they are confirmed or agree with the modeling analysis.

The docking pose and types of interaction were agreed with the experimental results of antibacterial activity of the main constituents of A. vera gel and CSNPs against H. pylori.

Conclusions

The main results of the study have been included in the conclusions. However, It is hard to review the quality of the conclusions because of the poor writing quality of the manuscript. Maybe including specific information within this section could improve the conclusions, e.g. the most relevant constituents responsible for the antioxidant characteristics of the gel; the quantitative difference in the bacteriostatic activity of the assayed materials against H. pylori; etc.

its reviewed

Reviewer 3 Report

Dear all, the paper has been revised. This work refers to the use of Aloe vera gel in application with chitosan nanoparticles. The work is presented in an interesting way, with a detailed description of the methodology.

I suggest the elaboration of a figure with a flowchart of the methodology, easily explaining the quality of the research, where the raw materials come from and the final use (product).

The results figure has a bad resolution, remove or improve the quality.

The fonts for the HPLC, FTIR and SEM figures are different and smaller. standardize them. Rectangular figures make the presentation "ugly". Make square figures.

Wouldn't it be possible to draw a figure from table 4?

Difficult to visualize in the figure on page 14, line 408. The proposal is interesting but difficult to understand.

I believe that after these corrections the paper can be accepted for publication.

Author Response

Dear all, the paper has been revised. This work refers to the use of Aloe vera gel in application with chitosan nanoparticles. The work is presented in an interesting way, with a detailed description of the methodology.

I suggest the elaboration of a figure with a flowchart of the methodology, easily explaining the quality of the research, where the raw materials come from and the final use (product).

The results figure has a bad resolution, remove or improve the quality.

Its cleared

The fonts for the HPLC, FTIR and SEM figures are different and smaller. standardize them. Rectangular figures make the presentation "ugly". Make square figures.

Wouldn't it be possible to draw a figure from table 4?

Its  drawn  

Difficult to visualize in the figure on page 14, line 408. The proposal is interesting but difficult to understand.

I believe that after these corrections the paper can be accepted for publication.

Round 2

Reviewer 1 Report

Dear authors,

I suggest putting figures near the first place they were mentioned in order to improve the following of the discussion.

Figures of lower sizes and resolution will be needed. For instance, FTIR can all be displayed in one picture.

Still missing the discussion of other works on chitosan NP and aloe vera extract....

Author Response

Reviewer 1: I suggest putting figures near the first place they were mentioned in order to improve the following of the discussion.

Response: Thank you sir, each figure under the its subtitle

Reviewer 1: Figures of lower sizes and resolution will be needed. For instance, FTIR can all be displayed in one picture.

Response: Thank you my prof., its corrected and some figures will delete after  accept correction from editor, another responce  if  in one figure its confused . In first revision three figure were in one figure , another reviewer request to separate the figure

Reviewer 1: Still missing the discussion of other works on chitosan NP and aloe vera extract....

Response: Thank you sir, the following citations were added in discussion

Antibacterial properties of A. vera inner gel were demonstrates against both susceptible and resistant H. pylori strains (Cellini et al., 2014). These results may resolve the multi-drug resistance phenomenon particularly in cases of H. pylori.

Furthermore, as proposed by Pandey and Mishra (2010), the inner gel of A. vera could be great effective when taken orally, because, in side living human as well as animal, both anthraquinones  and acemannans as phytoconstituents of A. vera  gel were able to guaranteeing its complete activity.

Sharifi-Rad,  et al. (2021) according to literature review, concluded that pre-clinical and clinical investigations promote the application of CSNPs in nanomedicine.

Effect of A. vera with CSNPs on inflammation and wound healing  were studied (Ranjbar  and Yousefi 2018), where A. vera gel was effective and becoming more effective when incorporated with CSNPs

Cellini L., Di Bartolomeo S., Di Campli E., Genovese S., Locatelli M. and Di Giulio M. 2014. In vitro activity of Aloe vera inner gel against Helicobacter pylori strains . Letters in Applied Microbiology. 59, 43-48. doi:10.1111/lam.12241

Pandey, R. and Mishra, A. (2010) Antibacterial activities of crude extract of Aloe barbadensis to clinically isolated bacterial pathogens. Appl Biochem Biotechnol 160, 1356– 1361.

Sharifi-Rad, J., Quispe, C., Butnariu, M. et al. Chitosan nanoparticles as a promising tool in nanomedicine with particular emphasis on oncological treatment. Cancer Cell Int 21, 318 (2021). https://doi.org/10.1186/s12935-021-02025-4

Ranjbar R, Yousefi A. Effects of Aloe Vera and Chitosan Nanoparticle Thin-Film Membranes on Wound Healing in Full Thickness Infected Wounds with Methicillin Resistant Staphylococcus Aureus. Bull Emerg Trauma. 2018 Jan;6(1):8-15. doi: 10.29252/beat-060102.

Reviewer 2 Report

Review Manuscript Number: polymers-1765013-R1

Title: Molecular Docking and Efficacy of Aloe vera gel Based on chitosan nanoparticles Against Helicobacter pylori: 

This submission is the first revision of an original manuscript related to the formulation of aloe vera gel with chitosan nanoparticles as a potential treatment for gastric infections of H. Pylori.  

Some of my comments and queries from the original version were properly addressed. Although, language is still an issue in the manuscript.

There are still many grammatical and spelling mistakes, and poorly structured sentences. 

For example. 

Line 134 - “Moreover, the docking study of the main components of A. vera gel on H. pylori.” The sentences has no verb or sense in English. I am guessing the authors tried to express something like: “Moreover, a docking study of the main components of A. Vera on H. Pylori was performed.”. 

I strongly suggest the authors hiring a professional service or a native speaker to revise the manuscript before submitting this type of articles. There are still too many hard-to-read, and even wrong paragraphs. 

Another examples:

Line 158-159 – “At 280 nm, the UV-detector was monitored.” The sentence makes little sense in English. I assume the authors wanted to say something like: “The UV-detector was set to 280 nm”.

Line 159-161 - “The qualitative detection of phenolic and flavonoids were detected according to…”, makes no sense, and maybe it should be rewritten to something like: “The qualitative detection of phenolic and flavonoids was 

performed according to…”

The whole paragraph of the section 2.4. is written with the same type of mistakes, and it is difficult to understand the content in the study. 

Besides of the language and structure of the manuscript some of my queries were not properly addressed.

Examples: 

For my question: 

I believe the authors performed the characterization of the products by TEM and FTIR with dry products. If so, do they expect to have the same distribution that the one in the aqueous suspension?

The answer was “To perform TEM and FTIR analysis, the sample must be dried”. That reply do not answer my question at all. 

For my question:

“How do you explain the size differences of CSNPs with and without A vera gel? It seems counterintuitive that the particles decrease in size when the gel is incorporated.” 

The answer was: under TEM, the size appeared reduced. That answer does not justify the observed results, and if it does, a proper discussion should be included or at least suggested for the readers. 

Author Response

Line 158-159 – “At 280 nm, the UV-detector was monitored.” The sentence makes little sense in English. I assume the authors wanted to say something like: “The UV-detector was set to 280 nm”.

Response: Thank you my prof. changed to :The UV-detector was set to 280 nm

Line 159-161 - “The qualitative detection of phenolic and flavonoids were detected according to…”, makes no sense, and maybe it should be rewritten to something like: “The qualitative detection of phenolic and flavonoids was 

performed according to…”

Response: Thank you my prof. changed to: The qualitative detection of phenolic and flavonoids were performed according to Abdelghany et al. (2021) compared to the injected standards of phenolic and flavonoids in HPLC

The whole paragraph of the section 2.4. is written with the same type of mistakes, and it is difficult to understand the content in the study. 

Response: Thank you my prof , its corrected : To obtain CSNPs -A. vera gel composite coating solution, A. vera gel was incorporated into CSNPs solution (stirred for 25 min. and then ultrasonicated for 45 min.) to obtain final concentration of A. vera gel concentrate/ CSNPs in the solution at 10% by weight (Torlak and Sert 2013).

Besides of the language and structure of the manuscript some of my queries were not properly addressed.

Examples: 

For my question: 

I believe the authors performed the characterization of the products by TEM and FTIR with dry products. If so, do they expect to have the same distribution that the one in the aqueous suspension?

The answer was “To perform TEM and FTIR analysis, the sample must be dried”. That reply do not answer my question at all. 

Response: No same distribution that the one in the aqueous suspension, therefore before all experimental the suspension was agitated

For my question:

“How do you explain the size differences of CSNPs with and without A vera gel? It seems counterintuitive that the particles decrease in size when the gel is incorporated.” The answer was: under TEM, the size appeared reduced. That answer does not justify the observed results, and if it does, a proper discussion should be included or at least suggested for the readers. 

Response: Deep very thanks my prof. very very thanks for this observation, I think size decreased as result of reaction among A vera gel content with CSNPs or may due to sonication before TEM Examination. therefore the sample was prepared two time and reexamined observe size

Reviewer 3 Report

All suggestions have been corrected. For my part, the paper can be accepted for publication.

Author Response

Reviewer 3

Comments and Suggestions for Authors

All suggestions have been corrected. For my part, the paper can be accepted for publication.

Response: Deep very thanks my prof.